# Connecting Neural Models Latent Geometries with Relative Geodesic Representations

**Hanlin Yu**[1]
University of Helsinki

**Berfin Inal**
University of Amsterdam

**Georgios Arvanitidis**
DTU

**Søren Hauberg**
DTU

**Francesco Locatello**
IST Austria

**Marco Fumero**[1]
IST Austria

## Abstract

Neural models learn representations of high-dimensional data on low-dimensional manifolds. Multiple factors, including stochasticities in the training process, model architectures, and additional inductive biases, may induce different representations, even when learning the same task on the same data. However, it has recently been shown that when a latent structure is shared between distinct latent spaces, relative distances between representations can be preserved, up to distortions. Building on this idea, we demonstrate that exploiting the differential-geometric structure of latent spaces of neural models, it is possible to capture *precisely* the transformations between representational spaces trained on similar data distributions. Specifically, we assume that distinct neural models parametrize approximately the same underlying manifold, and introduce a representation based on the *pullback metric* that captures the intrinsic structure of the latent space, while scaling efficiently to large models. We validate experimentally our method on model stitching and retrieval tasks, covering autoencoders and vision foundation discriminative models, across diverse architectures, datasets, pretraining schemes and modalities. Code is available at `https://github.com/marc0git/RelativeGeodesics`.

## 1   Introduction

Neural models learn meaningful representations of high-dimensional data generalizing to many tasks, spanning different data modalities and domains. Recent research reveals that these models often develop similar internal representations given similar inputs [Li et al., 2015, Moschella et al., 2023, Fumero et al., 2024, Kornblith et al., 2019], a phenomenon that was observed in biological networks [Laakso and Cottrell, 2000, Haxby et al., 2001]. Remarkably, even when models have different architectures, their internal representations can frequently be aligned through a simple, e.g. orthogonal, transformation [Maiorca et al., 2024, Lähner and Moeller, 2024, Moayeri et al., 2023]. This suggests a certain consistency in how neural nets encode information, emphasizing the importance of studying the

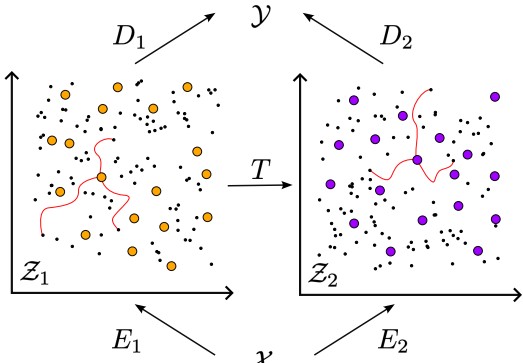

Figure 1: *Neural models trained on similar data learn parametrizations of the same manifold.* NNs learn parametrizations $(D_1, D_2)$ of the same underlying manifold $\mathcal{Y}$ up to isometries $T$. Pulling back the metric from $\mathcal{Y}$ makes relative geodesic representations invariant to transformations $T$ between latent spaces $\mathcal{Z}_1$ and $\mathcal{Z}_2$.

---

[1]Corresponding emails: `marco.fumero@ist.ac.at`, `hanlin.yu@helsinki.fi`

39th Conference on Neural Information Processing Systems (NeurIPS 2025).

internal representations and the transformations that relate them, to the extent to hypothesize whether neural nets are converging toward a unique representation of reality [Huh et al., 2024].

One strategy to understand how different models are related is to identify representations that are *invariant* to transformations between distinct models' representational spaces. A simple and effective recipe is that of *relative representations* [Moschella et al., 2023], where samples are represented as a function of a fixed set of latent representations. The similarity function employed is cosine similarity, hinting at the fact that representations across distinct models are subject to *angle preserving* transformations. However, the choice of similarity function should not be limited to only capturing invariances of one class of transformations. As shown in Cannistraci et al. [2024], Fumero et al. [2021], other choices can be good as well, and there is not a clear best choice among different transformations for capturing transformation across distinct latent spaces. We posit that when it is possible to relate distinct neural models' representational spaces, neural models are learning distinct parametrizations of the *same* underlying manifold (see Figure 1). In this paper, we employ geodesic distance in the latent space for relative representations. This approach ensures that the relative space remains approximately invariant to the isometries and reparametrization of the data's manifold, as characterized by a Riemannian structure. Our contributions can be summarized as follows:

- We observe that distinct neural models learn parametrization of the same underlying manifold when trained on similar data.

- We propose a new representation that captures the isometric transformation between data manifolds learned by distinct models, by leveraging the pullback metric.

- We propose to employ a scalable approximation of the geodesic energy to compute intrinsic distances that preserve the ranks of true distances.

- We show how to get meaningful pullback metrics from discriminative models, such as classifiers and self-supervised models.

- We test relative geodesics on retrieval and stitching tasks on autoencoders and vision foundation models, across different models, training schemes, and modalities, outperforming prior methods.

## 2 Related Work

**Representation alignment.** Numerous studies have shown that neural networks trained under different initializations, architectures, or objectives learn highly similar internal representations [Moschella et al., 2023, Bonheme and Grzes, 2022, Kornblith et al., 2019, Cannistraci et al., 2024, Li et al., 2015, Bengio et al., 2014, Maiorca et al., 2024, Huh et al., 2024, Guth et al., 2024, Chang et al., 2022, Conneau et al., 2018, Tsitsulin et al., 2020, Nejatbakhsh et al., 2024]. This correspondence becomes stronger in wide and large networks [Barannikov et al., 2022, Morcos et al., 2018, Somepalli et al., 2022]. Leveraging these aligned embeddings, a simple linear transformation often suffices to map one network's latent space onto another's, enabling techniques such as model stitching, where components from different networks can be interchanged with minimal loss in performance [Fumero et al., 2024, Bansal et al., 2021, Csiszárik et al., 2021]. In practice, aligning latent spaces using a linear transformation achieves comparable downstream task performance [Moayeri et al., 2023, Merullo et al., 2023, Maiorca et al., 2024, Lähner and Moeller, 2024].

**Latent space geometry.** Early work on the geometry of deep latent representations focused on autoencoders, where the decoder's mapping from latent to data space induces a natural *pullback metric* under the assumption that the ambient space is Euclidean [Shao et al., 2018, Tosi et al., 2014, Arvanitidis et al., 2018]. The Riemannian viewpoint allows one to compute geodesic paths and meaningful distances that respect the manifold structure of the learned embedding. Subsequent research has introduced computationally efficient approximations, such as energy-based proxies, and extended these ideas to estimate local curvature for improved interpolation and sampling [Chen et al., 2019, Chadebec and Allassonnière, 2022, Loaiza-Ganem et al., 2024, Arvanitidis et al., 2021, 2022a]. In the context of discriminative models, one can obtain a Riemannian metric primarily using two approaches [Grosse, 2022], either by pulling back the Fisher Information Matrix [Amari, 2016, Arvanitidis et al., 2022b] or by assuming a Euclidean geometry on the output space and pulling back the $L2$ metric. Interestingly, one can obtain some identifiability guarantees by taking geometry into consideration [Syrota et al., 2025].

# 3 Method

## 3.1 Notation and background

Neural networks (NNs) are parametric functions $F_\theta$, composed of an *encoding* map and a *decoding* map, represented as $F_\theta = D_{\theta_2} \circ E_{\theta_1}$. The encoder $E_{\theta_1} : \mathcal{X} \mapsto \mathcal{Z}$ generates a latent representation $z = E_{\theta_1}(x)$, where $x \in \mathcal{X}$ is mapped from the input domain $\mathcal{X}$ to the latent space $\mathcal{Z}$. The decoder $D_{\theta_2}$ is responsible for performing the task at hand, such as reconstruction or classification. For simplicity, we omit the parameter dependence $(\theta)$ in our notation moving forward. For any single module $E$ (or equivalently $D$), we use $E_\mathcal{X}$ to denote that the module $E$ was trained on the domain $\mathcal{X}$. In the next sections, we will provide the necessary background to introduce our method.

**Latent space communication.** Given a pair of domains $(\mathcal{X}, \mathcal{X}')$, a pair of neural models trained on them $(F_\mathcal{X}^1, F_{\mathcal{X}'}^2)$ and a partial correspondence between the domains $\Gamma : \mathcal{A}_\mathcal{X} \mapsto \mathcal{A}_{\mathcal{X}'}$ where $\mathcal{A}_\mathcal{X} \subset \mathcal{X}$ and $\mathcal{A}_{\mathcal{X}'} \subset \mathcal{X}'$, the problem of *latent space communication* is the one of finding a full correspondence $\Lambda : E^1(\mathcal{X}) \mapsto E^2(\mathcal{X}')$ between the two domains, from $\Gamma$. In a simplified setting, for example two models trained with different initialization or architectures on the same data, $\mathcal{X} = \mathcal{X}'$ and the correspondence is the identity. When $\mathcal{X} \neq \mathcal{X}'$ the problem recovers the multimodal setting.

**Relative representations.** The relative representations framework [Moschella et al., 2023] provides a straightforward approach to represent each sample in the latent space according to its similarity to a set of fixed training samples, denoted as *anchors*. Representing samples in the latent space as a function of the anchors corresponds to transitioning from an absolute coordinate frame into a *relative* one defined by the anchors and the similarity function. Given a domain $\mathcal{X}$, an encoding function $E_\mathcal{X} : \mathcal{X} \to \mathcal{Z}$, a set of anchors $\mathcal{A}_\mathcal{X} \subset \mathcal{X}$, and a similarity or distance function $d : \mathcal{Z} \times \mathcal{Z} \to \mathbb{R}$, the *relative representation* for a sample $x \in \mathcal{X}$ is:

$$RR(z; \mathcal{A}_\mathcal{X}, d) = \bigoplus_{a_i \in \mathcal{A}_\mathcal{X}} d(z, E_\mathcal{X}(a_i)),$$

where $z = E_\mathcal{X}(x)$, and $\bigoplus$ denotes row-wise concatenation. In the original method [Moschella et al., 2023], $d$ corresponds to cosine similarity. This choice induces a representation invariant to *angle-preserving transformations*. In this work, our focus is to *leverage the intrinsic geometry of latent spaces to employ a metric that captures isometric transformations between data manifolds*.

**Latent space geometry.** For the latent space of a neural network, it is generally hard to reason about its Riemannian structure. However, it is often easier to assign a Riemannian structure to the output space. As such, one can define a *pullback metric* from the output space to the latent space, which is a standard operation in Riemannian geometry (see Ch.2.4 of Do Carmo and Flaherty Francis [1992]).

Formally, the decoder $D : \mathcal{Z} \mapsto \mathcal{Y}$ takes as input a latent representation $z \in \mathcal{Z}$ and outputs $y$. Given a Riemannian metric defined on $y$ as $G_\mathcal{Y}(y)$, one can obtain the Riemannian metric at $z$ as:

$$G_\mathcal{Z}(z) = \left(\frac{\partial y}{\partial z}\right)^\top G_\mathcal{Y}(y) \left(\frac{\partial y}{\partial z}\right) = J_D(z)^\top G_\mathcal{Y}(y) J_D(z),$$

where $J_D(z)$ is the Jacobian of $D$ at $z$. The metric tensor $G_\mathcal{Y}$ is useful to compute quantities such as lengths, angles, and areas on $\mathcal{M}$. Given a smooth curve $\gamma : [a, b] \mapsto \mathcal{M}$, its arc length is defined as:

$$L(\gamma) = \int_a^b \sqrt{v(t)^\top G_\mathcal{Y}(\gamma(t)) v(t)^\top} \, dt, \tag{1}$$

where $v(t) = \dot{\gamma}(t)$. A slight variation of the above functional gives the geodesic energy $\mathcal{E}$ of $\gamma$ [Arvanitidis et al., 2018, Shao et al., 2018]

$$\mathcal{E}(\gamma) = \frac{1}{2} \int_a^b v(t)^\top G_\mathcal{Y}(\gamma(t)) v(t)^\top \, dt. \tag{2}$$

Both can be discretized and approximated in practice using finite difference approaches [Yang et al., 2018, Shao et al., 2018]. Geodesics minimize both the length and the energy, where for optimization the latter is usually preferred for numerical stability [Hauberg, 2025]. These quantities have the property of being *invariant* to certain reparametrizations, as formalized in the following proposition:

**Proposition 3.1.** *Let $\gamma : [0,1] \to \mathcal{M}$ be a smooth curve on a Riemannian manifold $(\mathcal{M}, G)$, and let $(\mathcal{M}', G')$ be a reparameterization of the manifold and $\varphi : [0,1] \to [0,1]$ a smooth, strictly increasing reparametrization of $\gamma$. Setting $\gamma'(\tau) = \gamma(\varphi(\tau))$ the Riemannian length and energy of $\gamma$ are invariant under reparameterizations of the manifold:*

$$\mathcal{E}[\gamma] = \frac{1}{2} \int_0^1 \left\| \tfrac{d\gamma'}{d\tau} \right\|_G^2 \, d\tau = \frac{1}{2} \int_0^1 \left\| \tfrac{d\gamma'}{d\tau} \right\|_{G'}^2 \, dt,$$

$$L[\gamma] = \int_0^1 \left\| \tfrac{d\gamma'}{d\tau} \right\|_G \, d\tau = \int_0^1 \left\| \tfrac{d\gamma'}{d\tau} \right\|_{G'} \, dt.$$

*Furthermore, the Riemannian arc length of $\gamma$ is invariant under reparametrizations $\gamma'$ on $\mathcal{M}$:*

$$L[\gamma'] = \int_0^1 \left\| \tfrac{d\gamma'}{d\tau} \right\|_G \, d\tau = \int_0^1 \left\| \tfrac{d\gamma}{dt} \right\|_G \, dt = L[\gamma].$$

We provide the proof in Appendix A.1.1.

## 3.2 Relative geodesics representations

---

**Algorithm 1** Relative Geodesic Representations

---

**Require:** Sample $x \in \mathcal{X}$, anchors $\mathcal{A}_\mathcal{X}$, encoder $E$, decoder $D$, distance $d_\mathcal{X}$ induced by metric $G_\mathcal{Y}$, steps $N$, step size $\Delta t$, mode $\in \{\texttt{energy}, \texttt{distance}\}$
**Ensure:** $RR^{geo}(x; \mathcal{A}_\mathcal{X})$
1: $z \leftarrow E(x), \quad RR^{geo} \leftarrow [\,]$
2: **for** $a \in \mathcal{A}_\mathcal{X}$ **do**
3: $\quad z_a \leftarrow E(a), \quad d \leftarrow 0$
4: $\quad$ **for** $j = 1$ to $N$ **do**
5: $\qquad \gamma_j \leftarrow (1 - \frac{j}{N})z + \frac{j}{N}z_a$
6: $\qquad \gamma_{j-1} \leftarrow (1 - \frac{j-1}{N})z + \frac{j-1}{N}z_a$
7: $\qquad v \leftarrow D(\gamma_j) - D(\gamma_{j-1})$
8: $\qquad G \leftarrow G_\mathcal{Y}\big(D(\gamma_j)\big)$
9: $\qquad s \leftarrow v^\top G v$
10: $\qquad d \leftarrow d + \Delta t \cdot (\texttt{energy} \Rightarrow \frac{1}{2}s, \texttt{distance} \Rightarrow \sqrt{s})$
11: $\quad$ **end for**
12: $\quad$ Append $d$ to $RR^{geo}$
13: **end for**
14: **return** $RR^{geo}$

---

From a differential geometry perspective, the problem of latent space communication can be interpreted as finding a transformation between the data manifolds $\mathcal{M}_1, \mathcal{M}_2$ approximated by two neural models $F_\mathcal{X}^1, F_{\mathcal{X}'}^2$. The relative representation framework captures this transformation implicitly if equipped with the right metric: we posit that a natural candidate for this metric is the geodesic distance defined on $\mathcal{M}_1, \mathcal{M}_2$, respectively. This choice makes the relative representations *invariant* to isometric transformation $T$ of the manifolds $\mathcal{M}_1, \mathcal{M}_2$. However, for high-dimensional problems, the high cost of computing the geodesic (corresponding to minimizing Eq. 2) makes this impractical [Shao et al., 2018, Chen et al., 2019]. Furthermore, one can argue against directly using the latent geometry induced by deterministic models from a theoretical perspective [Hauberg, 2019], as it may result in undesirable properties, for example the geodesics going outside of the data manifold.

We therefore approximate the geodesic quantities by directly considering the energy (or the length) of the straight line (in the Euclidean sense) connecting representations in the latent space:

$$RR^{geo}(z; \mathcal{A}_\mathcal{X}) = \bigoplus_{a_i \in \mathcal{A}_\mathcal{X}} \mathcal{E}(\tilde{\gamma}_\alpha(z, E_\mathcal{X}(a_i))),$$

where $\tilde{\gamma}_\alpha(z_1, z_2) = (1 - \alpha)z_1 + \alpha z_2$ is the convex combination between the points $z_1, z_2$. The approximation gives a natural upper bound to the geodesic distance: for $\bar{\gamma}$ it can be shown to relate to the arc length of a curve defined in Eq. 1 and the energy in Eq. 2 using the following bounds:

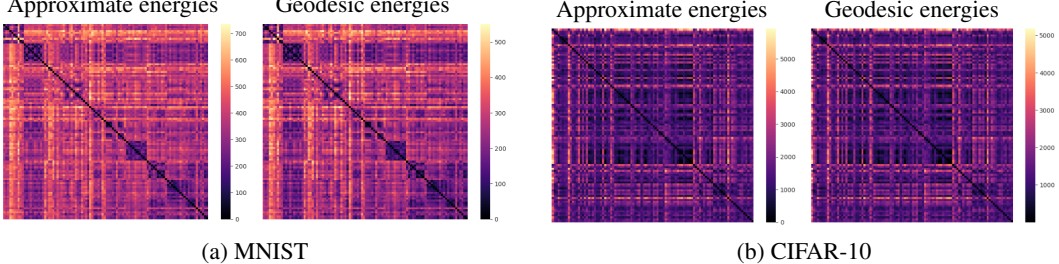

| Approximate energies | Geodesic energies | Approximate energies | Geodesic energies |

(a) MNIST (b) CIFAR-10

Figure 2: Pairwise latent-space energy matrices for (a) MNIST and (b) CIFAR-10. In each subfigure, the left heatmap shows the straight-line energy approximation and the right shows the geodesic energies of the ground truth geodesic curve. The Spearman rank correlations between the two measures are $\rho = 0.99$ for MNIST and $\rho = 1.00$ for CIFAR-10, demonstrating near-perfect agreements.

$$d(\boldsymbol{z}_0, \boldsymbol{z}_1)^2 \leq L^2(\tilde{\boldsymbol{\gamma}}) \leq 2\mathcal{E}(\tilde{\boldsymbol{\gamma}}). \tag{3}$$

The proof is in Appendix A.1.2. Moreover the approximation is far more *efficient* to compute, without requiring minimization of Equation 2, and is *accurate*, as empirically verified in Figure 2.

**Discretization.** When the step size is sufficiently small, the energy and arc length in the latent space, as defined in Equations 1 and 2, can be approximated by their counterparts in the output space using discretized finite difference schemes [Shao et al., 2018]:

$$\mathcal{E}(\boldsymbol{\gamma}) = \sum_{i=1}^{N} E_i = \frac{1}{2}\sum_{i=1}^{N} \boldsymbol{v}(t_i)^\top G(t_i)\boldsymbol{v}(t_i)\Delta t, \tag{4}$$

$$L(\boldsymbol{\gamma}) = \sum_{i=1}^{N} d_i = \sum_{i=1}^{N} \sqrt{\boldsymbol{v}(t_i)^\top G(t_i)\boldsymbol{v}(t_i)}\Delta t, \tag{5}$$

where $\Delta t = \frac{1}{N}$, with $N$ being the number of discretization steps. For Euclidean geometry, the geodesic arc lengths are given in closed form as the geodesics are straight lines. Unlike the energy, the curve length is invariant under reparametrizations (proposition 3.1). As such we focus on the curve length in our experiments. The resulting algorithm is summarized in Algorithm 1. In practice, with specific choice of $G_\mathcal{Y}$ one can avoid approximating the distance between $D(\boldsymbol{\gamma}_j, \boldsymbol{\gamma}_{j-1})$ explicitly using $G_\mathcal{Y}$ by directly calculating the distance or energy between $\boldsymbol{\gamma}_j$ and $\boldsymbol{\gamma}_{j-1}$ on $\mathcal{Y}$.

**Approximate geodesic energies.** Our choice comes with three advantages: *(i) efficiency*: avoiding minimization of Eq.2 the computation for every sample reduces to a single forward pass for every discretization step $\boldsymbol{\gamma}$ and for each anchor, resulting in overall complexity of $O(TA)$ forward passes of the decoder, where $A$ the number of anchors is the number of discretization steps. *(ii)* Directly using the arc length ensures *invariance* to reparametrizations of the manifold, matching our assumptions. *(iii)* As we only need reasonably accurate estimates of the arc lengths rather than the geodesic trajectory, the approach is *accurate*. Specifically, to assess how close the straight line energy approximation (2) is to the true geodesic energies, we first encoded 100 samples (10 per class, sorted by label) from MNIST [Deng, 2012] and CIFAR-10 using a simple convolutional autoencoder (architecture detailed in Appendix A.3.2). We then computed pairwise geodesic energy matrices over these latent representations using both methods, and the results are displayed in Fig. 2. Visually, both energy matrices exhibit the same block-diagonal structure, mainly due to belonging to the same class, and clustering patterns. Numerically, their Spearman rank correlation exceeds 0.99 with only 8 discretization points (see Appendix A.3.5 for correlation results across different numbers of discretization steps and for implementation details).

### 3.3 Choice of pullback metric

The properties of the relative geodesic representations are determined by (i) the choice of the output space, (ii) the choice of the metric to pullback from the output space and (iii) the pretraining objective (e.g. reconstruction or classification) on which the decoder was trained.

**Generative models.** For models trained on a reconstruction loss such as autoencoders, or on generative objectives, such as variational autoencoders Kingma and Welling [2013], pulling back metrics such as $L2$ distance have been shown to effectively reflect the underlying geometry of the latent space [Tosi et al., 2014, Arvanitidis et al., 2018, Hauberg, 2019].

**Discriminative models.** For discriminative models, such as classifiers or instance based discriminative models [Ibrahim et al., 2024], it is not immediate how to assign a Riemannian structure to the space of latent representations. From the perspective of information geometry, perhaps the most natural choice is the Fisher information matrix [Amari, 2016], in which case the metric in the output space can be obtained as the one with categorical likelihood. However, neural networks typically experience Neural Collapse [Kothapalli, 2023], possibly rendering the resulting geometry troublesome. We empirically inspect this approach in Appendix A.4.11. In this work we consider two principled approaches for discriminative models based on classification decoder heads and instance discrimination heads.

*Pulling back from classifiers.* Perhaps the most natural idea is, as discussed in Section 3.1, to construct a pullback metric based on the model's outputs, by simply pulling back the Euclidean $L2$ metric from the logit space of the classifier. Given an arbitrary encoder model, we train a classification head upon the latent representations (extracted e.g. from the last layer) and pulling back the Euclidean $L2$ metric from the output logits of the head. The resulting relative geodesics representation will inherit properties of both the decoder head (up to class information) and the pretrained encoder.

*Pulling back from instance discrimination decoders.* Diet [Ibrahim et al., 2024] is a self-supervised training method which has been shown to learn representations with strong generalization to downstream tasks, and yield identifiability guarantees [Reizinger et al., 2025]. Specifically, in the infinite data limit representations from the Diet objective align the cluster centers of von-Mises Fisher (vMF) distributions, which lie on a unit sphere. The loss based on simple instance discrimination is:

$$\mathcal{L}_{Diet} = \mathbb{E}_{\boldsymbol{x},i} \left[ -\log \frac{\exp\left(\boldsymbol{w}_i^\top f(\boldsymbol{x})\right)}{\sum_j \exp\left(\boldsymbol{w}_j^\top f(\boldsymbol{x})\right)} \right], \tag{6}$$

where $\boldsymbol{W}$ is a linear projection and $f$ is a nonlinear map. Further, assigning the same instance label to data augmentations was shown beneficial to improve invariance. Originally proposed to train the entire neural network [Ibrahim et al., 2024], we instead use it to learn a decoder $D$ on top of the pretrained neural network latent representations, by setting $D = f \circ \boldsymbol{W}$. To construct relative geodesic representations, we propose to pullback the spherical metric from the penultimate layer of the diet decoder, before $\boldsymbol{W}$. Further discussions on Diet can be found in Appendix A.2.1.

Both classifier and instance discriminator approaches discussed above use proper pullback metrics and fall under our proposed framework of relative geodesic representations: these representations inherit semantic information, up to class or instance level, from the decoder, while retaining structure of the pretrained encoder. Notably, when the encoder is pretrained, the relative geodesics representations are computationally efficient, since the decoder remains lightweight and the approximate geodesic energy computation is cheap. As we demonstrate in the following experimental sections, this approach yields meaningful and identifiable representations that are consistent across models.

## 4 Experiments

In the following we will evaluate the performance of relative geodesic representations on latent communication problems across models with different initializations, architectures, sizes and modalities.

**Tasks description.** We evaluate our approach on two representative instantiations of the latent communication problem: *retrieval* and *neural stitching*. In a retrieval setting we aim to solve the latent communication problem up to the *instance* level. Given pairs of model encoders $(E_\mathcal{X}^1, E_{\mathcal{X}'}^2)$ and access to their latent representations, we seek to recover a full correspondence $\Lambda$ starting from a partial one $\Gamma$. For neural stitching the goal is to solve the latent communication problem up to the *task-label* level. Classical stitching approaches train an adapter $\Psi$ between intermediate components of distinct neural networks so that $D_{\mathcal{X}'}^2 \circ \Psi \circ E_\mathcal{X}^1$ remains functional on a downstream task (e.g., classification). In Section 4.2, we operate in the *zero-shot stitching* regime [Moschella et al., 2023], where no adapter is trained explicitly. Instead, we solve implicitly for the transformations between representations by mapping them into *relative representation spaces*. This enables stitching pairs of models without any fine-tuning or additional supervision.

In Section 4.1, we evaluate *relative geodesic representations* on generative models, focusing on autoencoders. This analysis examines performance across networks trained with different initializations and datasets. In Section 4.2, we extend the evaluation to *discriminative foundation models*, assessing performance at scale across diverse architectures, pretraining objectives (e.g., self-supervised and classification), datasets, and modalities.

## 4.1 Experimental evaluation on autoencoders

In the following sections, we evaluate relative geodesic representations on the latent communication problem across autoencoder models trained with different initializations, architectures and datasets.

### 4.1.1 Aligning independently trained neural representational spaces

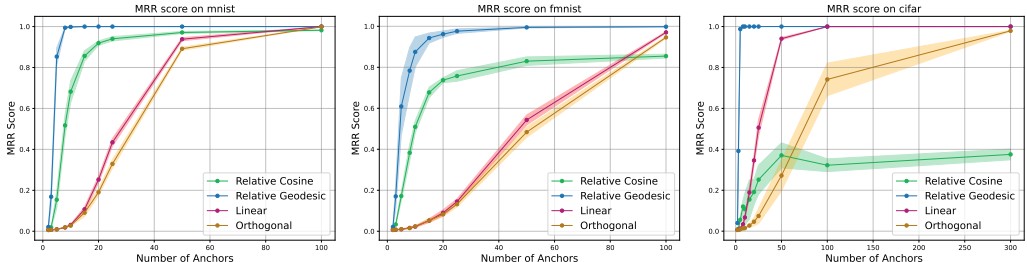

Figure 3: *Aligning latent spaces of autoencoders*: MRR score as a function of the number of anchors on pairs of autoencoders trained with different initializations on the `MNIST` (left), `FashionMNIST` (center), `CIFAR10` (right) datasets, respectively. In green, we plot the performance of Moschella et al. [2023]; in red and orange the linear and orthogonal baselines respectively; in blue, our method. The shaded area indicates standard deviation across 5 different random sets of anchors. Relative geodesic consistently outperforms baselines, obtaining peak performance.

**Setting.** For the following experiment, we trained pairs of convolutional autoencoders $(F_1, F_2)$ with different initializations on `MNIST` [Deng, 2012], `FashionMNIST` [Xiao et al., 2017], `CIFAR10` [Krizhevsky, 2009] datasets. The architecture of the convolutional autoencoder is detailed in Appendix A.3.2. After training, we extracted 10k samples from the test set, and mapped them to the latent spaces of the two models, to representations $\mathbf{Z}_1 = E_1(\mathbf{X}), \mathbf{Z}_2 = E_2(\mathbf{X})$ respectively. Starting from a small set of anchors in correspondence $\Gamma : \mathcal{A}_{\mathcal{X}} \mapsto \mathcal{A}_{\mathcal{Y}}$, the objective is to evaluate how well it is possible to recover the full correspondence $\Lambda$ between the representations $\mathbf{Z}_1, \mathbf{Z}_2$ from the relative representations. As a baseline, we compare with relative representations using cosine similarity [Moschella et al., 2023], and with fitting a linear or orthogonal mapping using $\Gamma$.

**Analysis of results.** Fig. 3 plots the performance in terms of MRR on `MNIST`, `FashionMNIST` and `CIFAR10` datasets. To obtain the score, we first compute similarity matrices between relative representations of the two spaces as $\mathbf{D}(\mathbf{Z}_1, \mathbf{Z}_2)$ where $\mathbf{D}_{i,j} = \frac{RR(\mathbf{Z}_1)_i^T RR(\mathbf{Z}_2)_j}{\|RR(\mathbf{Z}_1)_i\|_2 \|RR(\mathbf{Z}_2)_j\|_2}$. Then we compute the Mean Reciprocal Rank (MRR, see Appendix A.3.1) on top of the similarity matrix. In the figure, we plot MRR as a function of a random set of anchors, where the shaded areas indicate the standard deviations over 5 different sets of random anchors with the same cardinality. Our method consistently performs better than Moschella et al. [2023], saturating the score with few anchors on all the domains, despite the different degrees of complexity of the latent spaces. In addition, our method shows significantly less variance, being more robust to the choice of the anchor set.

**Takeaway.** Relative geodesic representation near-perfectly captures transformations between representational spaces of models initialized differently, sample efficiency and robustness.

### 4.1.2 Stitching autoencoder models

**Setting.** We consider the same pairs of autoencoders trained on the `MNIST`, `FashionMNIST`, `CIFAR10` datasets of Section 4.1.1. Starting from a set of five random anchors, we estimate a transformation $T$ between the model representational spaces $Z_1, Z_2$. In this experiment, to keep differently from Moschella et al. [2023], in which zero-shot stitching was achieved by training once a decoder

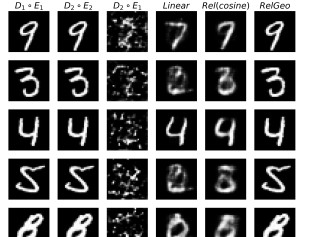 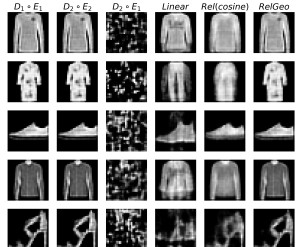 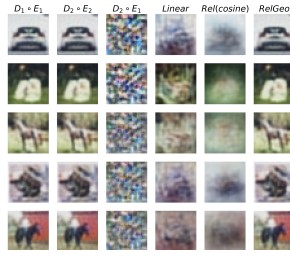

Figure 4: *Stitching on Autoencoders*: We visualize qualitative reconstructions of samples, stitching autoencoders of models trained with different initializations on `MNIST` (left), `FashionMNIST` (center), `CIFAR10` (right). The first two columns show reconstructions from the original models; middle three columns represent baselines [Maiorca et al., 2024, Moschella et al., 2023]; the rightmost column is our method. Relative geodesic yields the best stitching results using just 5 anchors.

Table 1: Average MRR cosine results for different methods across different datasets. Relative representations pulling back from diet decoder (`RelGeo(Diet)`) consistently provides better retrievals.

| Method | CIFAR-10 | CIFAR-100 | ImageNet-1k | CUB | SVHN |
|---|---|---|---|---|---|
| `Rel(Cosine)` [Moschella et al., 2023] | $0.129 \pm 0.135$ | $0.166 \pm 0.162$ | $0.221 \pm 0.178$ | $0.135 \pm 0.148$ | $0.068 \pm 0.08$ |
| `RelGeo($L2$)` | $0.047 \pm 0.013$ | $0.112 \pm 0.031$ | $0.412 \pm 0.09$ | $0.28 \pm 0.129$ | $0.025 \pm 0.012$ |
| `RelGeo(Diet)` | $\mathbf{0.387 \pm 0.145}$ | $\mathbf{0.445 \pm 0.142}$ | $\mathbf{0.566 \pm 0.111}$ | $\mathbf{0.523 \pm 0.177}$ | $\mathbf{0.314 \pm 0.188}$ |

module with relative representations and then exchanging different encoder modules, here we achieve stitching without training any decoder. We compute relative representations with respect to the set of anchors, and compute a similarity matrix $\mathbf{D}(\mathbf{Z}_1, \mathbf{Z}_2)$. Then we compute the vector $\mathbf{c} = \arg\max_i(\mathbf{D})$ representing a correspondence between the two representation matrices $\mathbf{Z}_1, \mathbf{Z}_2$, and use $c$ to fit a linear transformation $T$ to approximate the transformation between the two domains. We perform stitching by performing the following operation for a sample $x \in \mathcal{X}$: $\tilde{x} = D_2 \circ T \circ E_1(x)$.

**Analysis of results.** We visualize the results of reconstructions of random samples in Fig. 4, comparing against Moschella et al. [2023], Lähner and Moeller [2024], Maiorca et al. [2024]. For each dataset, each column represents respectively: (i) the original autoencoding mapping for a sample $x$ of model $F_1$, $D_1(E_1(x))$, (ii) $D_2(E_2(x))$, (iii) the mapping $D_2(E_1(x))$, (iv) the mapping $D_2(T_{anchors}E_1(x))$ where $T_{anchors}$ is estimated on the five available anchors, (v) the mapping $D_2(T_{cosine}E_1(x))$ where $T_{cosine}$ is estimated among all 10k samples with the correspondence $c$ obtaining in the relative space of Moschella et al. [2023], (vi) Our result $D_2(T_{relgeo}E_1(x))$, where $T_{relgeo}$ is estimated from the correspondence obtained in the relative geodesic space. As shown in Fig. 4, while the baselines do not reach a good enough reconstruction quality, reconstructions with our method are almost perfect in accordance with the results in Fig. 3.

**Takeaway.** Relative geodesics enable stitching of neural modules trained with different initializations.

## 4.2 Experiments on vision foundation models

In this section, we evaluate relative geodesic representations' performances on retrieval and model stitching tasks on vision foundation discriminative models across models pretrained with different objectives, architectures, sizes and modalities.

### 4.2.1 Matching representational spaces of discriminative foundation models

In this section, we test the compatibilities of representations of vision foundation models with different architectures, such as residual networks [He et al., 2016] and vision transformers [Dosovitskiy et al., 2021], and with different pretraining objectives including classification and self-supervised learning.

**Setting.** We perform experiments on retrieval tasks on pretrained vision foundation models, investigating how well we can match representations together with different backbones subject to the decoding tasks, on 5 datasets, varying in complexity and size: `CIFAR10`, `CIFAR100` [Krizhevsky, 2009], `SVHN` [Yuval Netzer et al., 2011], `CUB` [Wah et al., 2023], and `ImageNet-1k` [Russakovsky et al., 2015]. For ImageNet-1k, we used 1000 anchors, while for other datasets we used 500. As

backbones we consider ResNet-50 [He et al., 2016], Vision Transformers (ViT) [Dosovitskiy et al., 2021] with both patch 16-224 and patch 32-384, and DINOv2 [Oquab et al., 2024]. We compare the original formulation of relative representations with cosine similarity [Moschella et al., 2023] denoted as `Rel(Cosine)`, relative geodesic representations pulling back from Euclidean logits denoted as `RelGeo(L2)`, and pulling back the spherical metric using a Diet decoder denoted as `RelGeo(Diet)`.

**Analysis of results.** Table 1 shows results from different methods averaged across all possible pairs of models on the considered datasets. Additionally, Fig. 5 shows the results on CUB. While `RelGeo(L2)` may result in worse MRR numbers, `RelGeo(Diet)` provides consistently improved retrieval performance. In Appendix A.4.1 we report full results for the datasets.

**Takeaway.** Relative geodesic representations pulling back from instance discrimination decoders are identifiable across vision foundation models, improving retrieval performances.

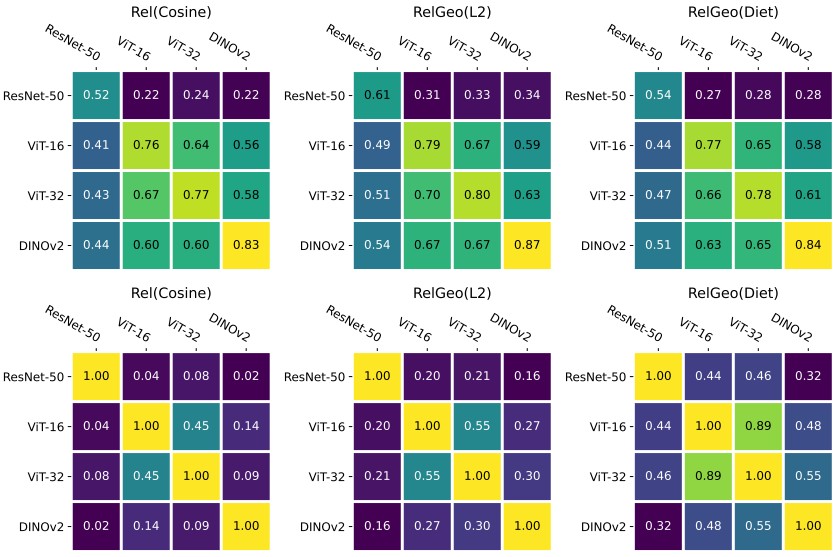

Figure 5: CUB Accuracies (top) and symmetricized MRR cosine (bottom). `RelGeo(Diet)` and especially `RelGeo(L2)` provide strong stitching accuracies, while `RelGeo(Diet)` maintains strong instance identifiability.

#### 4.2.2 Zero-shot stitching of vision foundation models

Table 2: Average stitching performances across different settings. `RelGeo(L2)` often outperforms `Rel(Cosine)`, while `RelGeo(Diet)` remains competitive.

| Method | CIFAR-10 | CIFAR-100 | ImageNet-1k | CUB | SVHN |
|---|---|---|---|---|---|
| `Rel(Cosine)` [Moschella et al., 2023] | $0.907 \pm 0.09$ | $0.775 \pm 0.132$ | $\mathbf{0.549} \pm 0.152$ | $0.531 \pm 0.188$ | $0.384 \pm 0.115$ |
| `RelGeo(L2)` | $\mathbf{0.955} \pm 0.03$ | $\mathbf{0.874} \pm 0.055$ | $0.501 \pm 0.159$ | $\mathbf{0.595} \pm 0.163$ | $\mathbf{0.59} \pm 0.054$ |
| `RelGeo(Diet)` | $0.915 \pm 0.074$ | $0.775 \pm 0.115$ | $0.479 \pm 0.17$ | $0.559 \pm 0.171$ | $0.416 \pm 0.079$ |

Model stitching was introduced in Lenc and Vedaldi [2015] to analyze neural network representational spaces, by training a linear layer to connect different layers and evaluating performance. Here we sidestep the need for trainable stitching layers and consider the zero-shot model stitching task defined in Moschella et al. [2023] to effectively test how components of vision foundation models can be reused. To do this, we leverage the space of relative geodesic representations as a shared compatible space. For the $i$th model $E_i$, we train one decoder $D_i$ on the relative representations induced by it, then evaluate the performance of using $D_i$ to decode the representations of model $E_j$, where $E_j$ may be a different model. This assesses how much two representation spaces can be merged with respect to the task defined by the decoder $D$, e.g., a classification head.

**Setting.** We perform experiments on pretrained vision foundation models from Hugging Face Transformers [Wolf et al., 2020], investigating how well we can match representations together for

classification with different backbones with classification heads, on the same datasets and models as considered in Section 4.2.1, similarly comparing `Rel(Cosine)`, `RelGeo(L2)` and `RelGeo(Diet)`.

**Analysis of results.** The results of the different methods across the different datasets are shown in Table 2, where we average over all possible model pairs. We further show the accuracies of the models on the CUB dataset in Fig. 5. Both `RelGeo(L2)` and `RelGeo(Diet)` provide strong stitching accuracies, with `RelGeo(L2)` reflecting the benefits of pulling back class specific information. `RelGeo(Diet)` still results in good accuracies while having very strong MRR metrics, as shown in 1.

**Takeaway.** Relative geodesic representations yield good accuracies and good MRRs, avoiding downgrading of performance when performing model stitching while retaining sample identifiability.

### 4.2.3 Matching different modalities

In this section we evaluate relative geodesic representations in the multimodal setting.

**Setting.** We study the retrieval task in terms of vision foundation models and the text encoders of CLIP [Radford et al., 2021] with both patch 16 and patch 32, using Flickr30k dataset [Young et al., 2014]. Keeping the text encoder of CLIP fixed, we swap the vision encoder with the ones of ResNet-50, DINOv2 and ViT, including different patch and model sizes. Due to the lack of class labels, `RelGeo(L2)` is not applicable, and we compare `RelGeo(Diet)` with `Rel(Cosine)`. While we observed that using data augmentations is beneficial for `RelGeo(Diet)`, due to the lack of a principled approach to construct data aug-

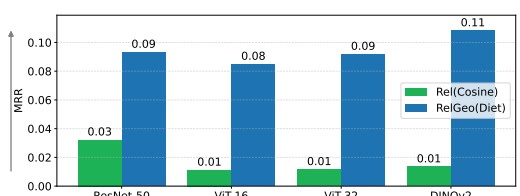

Figure 6: *Matching multimodal models.* Symmetricized MRR cosine on Flickr30k. RelGeo(Diet) substantially improves upon Rel(Cosine) in aligning multimodal models.

mentations on texts corresponding to image augmentations, we do not employ augmentations.

**Analysis of results.** The results in terms of symmetricized MRR cosine metric with CLIP with patch 16 are shown in Figure 6. We observe that RelGeo(Diet) yields significantly improved stitching performances upon Rel(Cosine). In Appendix A.4.9 we show the full pairwise matrices of MRR, comprising of the unimodal performances, inter vision models, and text models.

**Takeaway.** Relative geodesic representations show promising results for obtaining identifiable representations in multimodal scenarios.

## 5 Conclusions and discussion

We have introduced the framework of relative geodesic representation starting from the assumption that distinct neural models trained on similar data distributions learn to approximate the same underlying latent manifold. As a result, geodesic distances based on their representations are invariant to transformations between different representational spaces. We show that the geodesic energy and arc length of straight lines provide an efficient, low-cost metric for bridging these spaces, allowing us to measure similarity and align representations across different architectures, training objectives, and training procedures, while outperforming previous methods.

**Limitations and future work.** The accuracy of approximating geodesics using straight-line arc length (or energy) can deteriorate in regions of high curvature in the latent space, typically corresponding to areas far from the support of the training data. Moreover, this could require increasingly smaller step sizes, hurting the efficiency performance of the method. This suggests exploring nonlinear paths, and adaptive step sizes, e.g., by estimating the support of the data building KNN graphs in the latent space and forcing the path to not deviate too much from them. By employing the pullback metric from a given output space, the relative geodesic representation has the interesting property of restricting the alignment problem to the information relevant to the decoding task. This could be useful to *(i)* further explore no training multi-modal alignment [Norelli et al., 2023], where it is of interest to capture not only the shared information across modalities, but also the modality-specific information; *(ii)* to better understand the relation between the representation similarity and decodability [Harvey et al., 2024] and the interaction between tasks and learned representations [Fumero et al., 2023].

## Acknowledgments and Disclosure of Funding

We thank Gregor Krzmanc, German Magai, Vital Fernandez for insightful discussions in the early stages of the project. HY was supported by the Research Council of Finland Flagship programme: Finnish Center for Artificial Intelligence FCAI. HY wishes to acknowledge CSC - IT Center for Science, Finland, for computational resources. GA was supported by the DFF Sapere Aude Starting Grant "GADL". SH was supported by a research grant (42062) from VILLUM FONDEN and partly funded by the Novo Nordisk Foundation through the Center for Basic Research in Life Science (NNF20OC0062606). SH received funding from the European Research Council (ERC) under the European Union's Horizon Programme (grant agreement 101125003). MF is supported by the MSCA IST-Bridge fellowship which has received funding from the European Union's Horizon 2020 research and innovation program under the Marie Skłodowska-Curie grant agreement No 101034413.

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

# A Appendix

## A.1 Proof of theoretical results

### A.1.1 Proof of Proposition 3.1

*Proof.* We first prove the first half, i.e. the invariance of Riemannian curve length and energy across reparameterizations of the manifold. This can be proven by observing that the inner product at a point along the curve is invariant across such reparameterizations:

$$\|\dot{\boldsymbol{x}}\|_G = \dot{\boldsymbol{x}}^\top G(\boldsymbol{x})\dot{\boldsymbol{x}} = \left(\tfrac{d\boldsymbol{x}}{d\boldsymbol{x}'}\dot{\boldsymbol{x}'}\right)^\top \left(\tfrac{d\boldsymbol{x}'}{d\boldsymbol{x}}\right)^\top G'(\boldsymbol{x}')\tfrac{d\boldsymbol{x}'}{d\boldsymbol{x}}\tfrac{d\boldsymbol{x}}{d\boldsymbol{y}}\dot{\boldsymbol{x}'}$$
$$= \dot{\boldsymbol{x}'}^\top G'(\boldsymbol{x}')\dot{\boldsymbol{x}'} = \|\dot{\boldsymbol{x}'}\|_{G'}.$$

As such, the length and the energy of the same curves on different manifolds are integrals of the same quantities, hence are equal.

We then prove the second half, i.e. the invariance of Riemannian curve length across reparameterizations of the curve. Based on Equation 4.7 from Hauberg [2025], we have

$$L[\boldsymbol{\gamma}'] = \int_0^1 \left\|\tfrac{d\boldsymbol{\gamma}}{d\tau}\right\|_G d\tau = \int_0^1 \left\|\tfrac{d\boldsymbol{\gamma}}{d\tau}\right\|_G \tfrac{\varphi(t)}{t}\, dt$$
$$= \int_0^1 \left\|\tfrac{d\boldsymbol{\gamma}}{d\tau}\tfrac{\varphi(t)}{t}\right\|_G dt = \int_0^1 \left\|\tfrac{d\boldsymbol{\gamma}}{dt}\right\|_G dt$$
$$= L[\boldsymbol{\gamma}].$$

$\square$

### A.1.2 Proof of Equation 3

*Proof.* We first prove $d(\boldsymbol{z}_0, \boldsymbol{z}_1)^2 \leq L^2(\tilde{\boldsymbol{\gamma}})$, then prove $L^2(\tilde{\boldsymbol{\gamma}}) \leq 2\mathcal{E}(\tilde{\boldsymbol{\gamma}})$.

For the first part, according to the definition of geodesic distance, we have $d(\boldsymbol{z}_0, \boldsymbol{z}_1) \leq L(\tilde{\boldsymbol{\gamma}})$ and, as such, $d(\boldsymbol{z}_0, \boldsymbol{z}_1)^2 \leq L^2(\tilde{\boldsymbol{\gamma}})$.

The second part involves $L^2(\tilde{\boldsymbol{\gamma}}) \leq 2\mathcal{E}(\tilde{\boldsymbol{\gamma}})$, which can be proven using the Cauchy-Schwarz inequality. See Equation 7.14 in [Hauberg, 2025], where we denote $\boldsymbol{u}_t = \tfrac{d\tilde{\boldsymbol{\gamma}}}{dt}$ and $\boldsymbol{v}_t = 1$.

$$L^2(\tilde{\boldsymbol{\gamma}}) = \int_0^1 \left\|\tfrac{d\tilde{\boldsymbol{\gamma}}}{dt}\right\| dt = \langle \boldsymbol{u}, \boldsymbol{v}\rangle \leq \|\boldsymbol{u}\|\|\boldsymbol{v}\| \quad = \sqrt{\int_0^1 \left\|\tfrac{d\tilde{\boldsymbol{\gamma}}}{dt}\right\|^2 dt}\sqrt{\int_0^1 1^2\, dt} = \sqrt{\int_0^1 \left\|\tfrac{d\tilde{\boldsymbol{\gamma}}}{dt}\right\|^2 dt}$$
$$= 2\mathcal{E}(\tilde{\boldsymbol{\gamma}}).$$

$\square$

## A.2 Additional explanations

### A.2.1 Details on Diet

Proposed as a self-supervised learning method [Ibrahim et al., 2024], Diet was also shown to yield interesting identifiable guarantees [Reizinger et al., 2025], laying the theoretical foundation for `RelGeo(Diet)`, where we employ the resulting geometry.

One can consider such a scenario [Reizinger et al., 2025]: some latent variables $\boldsymbol{z}$ are drawn from a vMF distribution, and pushed forward through a continuous and injective generator function $g$ to obtain the data $\boldsymbol{x}$. Remarkably, given only $\boldsymbol{x}$ without the knowledge of $g$, it is possible to (to some degree) recover the latent variables $\boldsymbol{z}$ through parameterizing a model and optimizing the instance discrimination loss as given in Equation 6. Specifically, suppose there is a finite set of vectors $\boldsymbol{v}_c$ on a unit sphere, each representing a class, and a finite set of instances. One instance belongs to

exactly one class, and every class is employed by some instance. Additionally, the instance labels are chosen uniformly, and the latent variables $\boldsymbol{z}$ are drawn from a vMF distribution centered around the corresponding cluster vector $\boldsymbol{v}_c$ with concentration parameter $\kappa$.

Then, after the model is trained using the loss function as in Equation 6, when both $f$ and $w$ are not unit-normalized, $f \circ g$ is linear. This can be proven rigorously by expanding upon the theoretical framework of non-linear ICA [Hyvärinen et al., 2023]. As such, we propose to utilize $f \circ g$ to form the representations. For further technical details on the assumptions and additional results, we refer interested readers to Reizinger et al. [2025].

Assuming spherical geometry, the distance between two points $\boldsymbol{x}$ and $\boldsymbol{y}$ can be computed as

$$d(\boldsymbol{x}, \boldsymbol{y}) = \arccos \left( \frac{\boldsymbol{x}^\top \boldsymbol{y}}{\|\boldsymbol{x}\| \, \|\boldsymbol{y}\|} \right).$$

In the above formula, points that do not precisely lie on the unit sphere are effectively projected onto it. Interestingly this bears a strong resemblance to the cosine distances as used in the original paper on relative representations [Moschella et al., 2023].

### A.2.2 Why it works

Prior work on representational alignment has shown that representations from different models can often be approximately aligned using simple transformations, e.g. linear, orthogonal or locally linear maps. Even when models are trained independently – with differrent architectures, modalities, or datasets that nonetheless share an underlying structure – they tend to learn similar representations, suggesting convergence towards a shared encoding of entities [Huh et al., 2024]. From a theoretical standpoint, identifiability results [Roeder et al., 2021] imply that if two discriminative models learn the same likelihood function, their internal representations must be equivalent up to a linear transformation. However, this ideal scenario rarely holds exactly in practice: training dynamics, nuisance factors, and unmodeled variability can all introduce distortion. In our case, it may be too strong to assume that two models learn different parameterizations of an identical manifold. Instead, we adopt a weaker assumption, that they do so up to some bounded distortion. Recent theoretical work has begun to explore relaxations of strict identifiability to account for such bounded distortions [Nielsen et al., 2025]. Integrating these relaxations into our Riemannian framework presents a promising direction for future work.

In general, the few theoretical results available, e.g. [Roeder et al., 2021], often rely on unrealistic assumptions, e.g. proofs in axiomatic settings, infinite-data regimes, or the requirement that two models learn exactly the same likelihood function. In our view, a meaningful first step toward bringing theory and practice is to relax these assumptions, as initiated in [Nielsen et al., 2025], and begin to model more realistic scenarios, e.g. including the dynamics introduced by model training.

We believe that Riemannian methods can play a key role in this direction to try to capture local alignments beyond linear global transformations of the space as considered in Roeder et al. [2021] and possibly accounting for distortions measured in the linear space in practice. Nevertheless, we remark that neural networks could find qualitatively different solutions [Pascanu et al., 2025] and that the union of manifolds hypothesis might be more appropriate for modeling image data [Brown et al., 2023].

### A.3 Additional details

### A.3.1 Mean Reciprocal Rank

Mean Reciprocal Rank (MRR) is a commonly used metric to evaluate the performance of retrieval systems, and has been used to evaluate the capabilities of representations for instance discrimination [Moschella et al., 2023]. It measures the effectiveness of a system by calculating the rank of the first relevant item in the search results for each query.

To compute MRR, we consider the following steps:

1. For each query, rank the list of retrieved items based on their relevance to the query.
2. Determine the rank position of the first relevant item in the list. If the first relevant item for query $i$ is found at rank position $r_i$, then the reciprocal rank for that query is $\frac{1}{r_i}$.

3. Calculate the mean of the reciprocal ranks over all queries. If there are $Q$ queries, the MRR is given by:

$$\text{MRR} = \frac{1}{Q} \sum_{i=1}^{Q} \frac{1}{r_i}.$$

Here, $r_i$ is the rank position of the first relevant item for the $i$-th query. If a query has no relevant items in the retrieved list, its reciprocal rank is considered to be zero.

MRR provides a single metric that reflects the average performance of the retrieval system, with higher MRR values indicating better performance.

Similar to stitching accuracies, MRR is generally asymmetric. However, it can also be made symmetric. Specifically, as MRR is calculated based on a distance matrix $D$, one can make the distance matrix symmetric by setting $D = \frac{1}{2}\left(D^\top + D\right)$. In Section 4.2.1 we reported the symmetric version. Otherwise we report both the original version and the symmetric version, and discriminate between these two by explicitly indicating it when it is symmetric.

### A.3.2 Architectural details

We provide in Table 3 the architectural details of the convolutional autoencoders employed in experiments in Figures 3 and 4.

Table 3: Architecture of the convolutional autoencoders.

| Encoder |
| --- |
| $3 \times 3$ conv. 32 stride 2-ReLu |
| $3 \times 3$ conv. 64 stride 2-ReLu |
| Flatten |
| $(64 * k * k) \times h$ Linear |
| Latents |

| Decoder |
| --- |
| $h \times (64 * k * k)$ Linear |
| Unflatten |
| $3 \times 3$ conv. 64 stride 2-ReLu |
| $3 \times 3$ conv. 32 stride 2-ReLu |
| Sigmoid |

For the classifier experiment, in order to obtain geometric representations we need a decoder. The architecture is shown in Table 4. For RelGeo(Diet), the last linear layer is configured with *bias=False* in accordance with the original algorithm.

For evaluating the performances of the representations, we train a classification head with the same architecture as used by Moschella et al. [2023] as given in Table 5.

Table 4: Architecture of the simple decoders.

| Classification head |
| --- |
| $input\_dim$ LayerNorm |
| $input\_dim \times 500$ Linear-Tanh |
| $500 \times num\_classes$ Linear |

Table 5: Architecture of the decoders for evaluations.

| Final classification head |
| --- |
| $input\_dim$ LayerNorm |
| $input\_dim \times input\_dim$ Linear-Tanh |
| InstanceNorm1d |
| $input\_dim \times num\_classes$ Linear |

### A.3.3 RelGeo(Diet) augmentations

As noted by Ibrahim et al. [2024], it is beneficial to employ data augmentations when using Diet to perform self-supervised training of neural networks. We largely follow their approach, and considered different levels of data augmentations. Following Ibrahim et al. [2024], we consider different levels

of data augmentations indexed by a scalar strength, which are summarized below using PyTorch pseudocode; strengths of a higher level employs the augmentations of lower levels as well.

0: No augmentations;

1: RandomResizedCrop((height, width)), RandomHorizontalFlip();

2: RandomApply(ColorJitter(0.4, 0.4, 0.4, 0.2)), p=0.3); RandomGrayscale(0.2);

3: RandomApply(GaussianBlur((3, 3), (1.0, 2.0)), p=0.2), RandomErasing(0.25).

### A.3.4 Compute resources

Experiments regarding the geodesic approximation are conducted using NVIDIA A100 GPU and 12 CPU cores. Run time varies depending on the discretization steps, number of anchors and the used dataset.

The autoencoder stitching and retrieval experiments were conducted on a single NVIDIA RTX 3080TI GPU. Experiments involving vision foundation models were run on a compute cluster, each job using a single NVIDIA A100 GPU and 10 CPU cores, with runtimes of several hours. Preliminary experiments required additional resources, and in total we estimate having used several hundred GPU hours.

Further ablation studies on the running times can be found in Section A.4.10.

### A.3.5 Geodesic approximation

Here, we provide the experimental details of the results presented in Fig. 2 and Fig. 7. To assess the geodesic energies, we used a small autoencoder, whose architecture is presented in Table 6.

**Autoencoder training**   We trained a lightweight convolutional autoencoder (see Table 6) on both MNIST and CIFAR-10 to obtain the latent representations used in our experiments. For MNIST, the first convolutional layer was adjusted to accept a single input channel; for CIFAR-10 it used three channels. Each model was trained for 30 epochs using the Adam optimizer [Kingma and Ba, 2017] with a batch size of 64. We set the learning rate to 0.001, and fixed a random seed of 42 to ensure reproducibility.

**Energy computation**   After training, we selected 10 samples per class (100 total) in label order from each dataset and encoded them to produce their latent encodings. True geodesics are computed using Stochman library [Detlefsen et al., 2021], which has Apache-2.0 license, which wraps the decoder into a pullback manifold, intializes a parameterized spline path between codes, and then optimizes its parameters to minimize the Riemannian energy. Geodesic energies are computed as in Eq. 2. Pairwise energies are computed and visualized in Figures 2 and 7, demonstrating the close agreement between the two measures under identical encoding and discretization settings. In Fig. 2, latent dimensions for MNIST and CIFAR are 64 and 128 respectively, while in Fig. 7, latent dimension is 2 for both datasets.

Table 6: ConvAutoencoder architecture (latent dim $d$).

| Encoder | Activation |
|---|---|
| Conv2d(1, 32, kernel = 3, stride=2, pad=1) | ReLU |
| Conv2d(32, 64, kernel = 3, stride=2, pad=1) | ReLU |
| Flatten | — |
| Linear(64*7*7, $d$) | — |

| Decoder | Activation |
|---|---|
| Linear($d$, 64*7*7) | ReLU |
| Unflatten(64,7,7) | — |
| ConvTranspose2d(64, 32, kernel = 3, stride=2, pad=1, out_pad=1) | ReLU |
| ConvTranspose2d(32, 1, kernel = 3, stride=2, pad=1, out_pad=1) | Sigmoid |

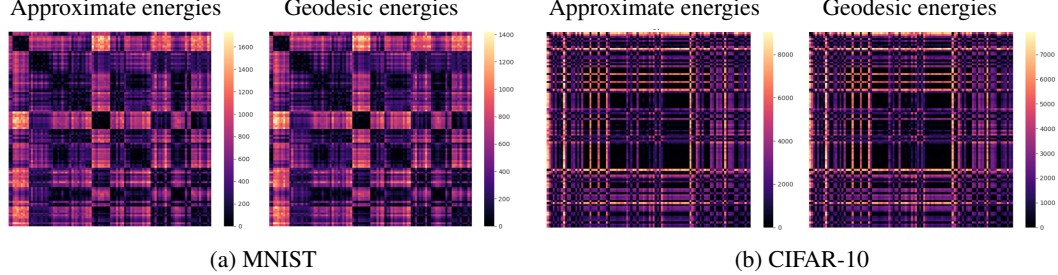

|  (a) MNIST  |  (b) CIFAR-10  |

Figure 7: Pairwise latent-space energy matrices for (a) MNIST and (b) CIFAR-10, with latent dimensionality 2. In each subfigure, the left heatmap shows the straight-line energy proxy and the right shows the full Riemannian geodesic energies. The Spearman rank correlation between the two measures is 0.99 for MNIST and $\rho = 1.00$ for CIFAR-10, demonstrating near-perfect agreement.

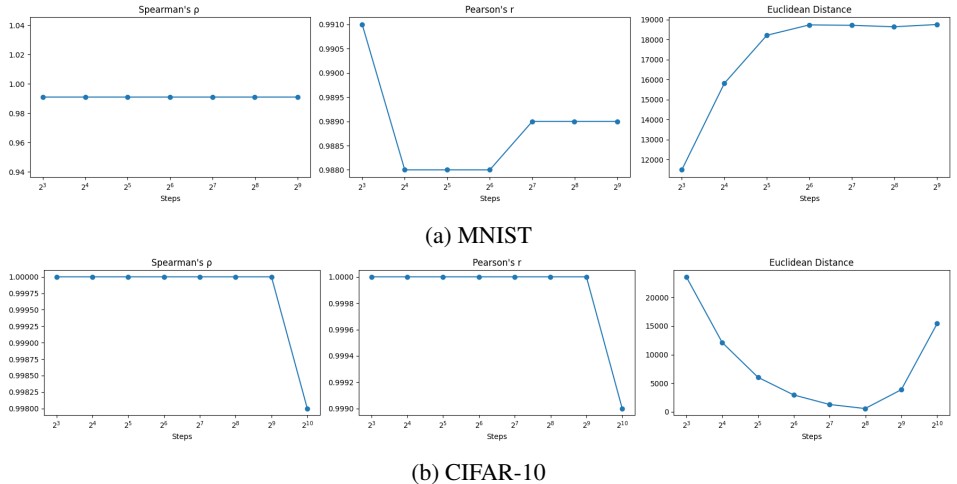

(a) MNIST

(b) CIFAR-10

Figure 8: Impact of varying discretization levels on similarity and energy metrics for (a) MNIST and (b) CIFAR-10 datasets. Each subplot shows how Spearman's $\rho$, Pearson's $r$, and Euclidean distance change as the number of discretization levels increases.

### A.3.6 Autoencoder stitching and retrieval

We provide the experimental details of the results presented in Figure 3 and Figure 4. All models employed followed the architecture depicted in Table 6, with a latent dimensionality of 128.

We trained the lightweight convolutional autoencoder (see Table 6) on MNIST, CIFAR-10, FashionM-NIST with 5 different seeds, to obtain the latent representations used in our experiments. For MNIST and FashionMNIST the first convolutional layer was adjusted to accept a single input channel; for CIFAR-10 it used three channels. Each model was trained for 50 epochs, reaching convergence, using the Adam optimizer [Kingma and Ba, 2017] with a batch size of 64. We set the learning rate to 0.001.

### A.3.7 Vision foundation models

We use the pretrained models as provided by Huggingface Transformers [Wolf et al., 2020], which has Apache-2.0 license, and the datasets as provided by HuggingFace Datasets [Lhoest et al., 2021], which also has Apache-2.0 license. The license information of the datasets are: CIFAR-10: unknown; CIFAR-100: unknown; CUB: unknown; ImageNet-1k: ImageNet agreement; SVHN: non-commercial use only.

Unless otherwise stated, we directly use the original test set of the dataset as the test set, while using 0.9 of the original train set as the train set and the remaining as the validation set. Both the anchors and the Diet data points are selected from the validation set.

For CIFAR-100, we use the coarse labels. For SVHN, the objective is to predict the cropped digits. For CUB dataset, we use the version available at `https://huggingface.co/datasets/`

Table 7: Aggregated results of MRR CDist Sym.

| Method | CIFAR-10 | CIFAR-100 | ImageNet-1k | CUB | SVHN |
|---|---|---|---|---|---|
| Rel(Cosine) [Moschella et al., 2023] | $0.098 \pm 0.133$ | $0.122 \pm 0.164$ | $0.103 \pm 0.146$ | $0.046 \pm 0.055$ | $0.046 \pm 0.081$ |
| RelGeo(L2) | $0.046 \pm 0.013$ | $0.105 \pm 0.031$ | $0.179 \pm 0.173$ | $0.187 \pm 0.141$ | $0.04 \pm 0.021$ |
| RelGeo(Diet) | $\mathbf{0.252} \pm 0.189$ | $\mathbf{0.278} \pm 0.211$ | $\mathbf{0.462} \pm 0.148$ | $\mathbf{0.433} \pm 0.212$ | $\mathbf{0.306} \pm 0.188$ |

Table 8: Aggregated results of MRR Cosine.

| Method | CIFAR-10 | CIFAR-100 | ImageNet-1k | CUB | SVHN |
|---|---|---|---|---|---|
| Rel(Cosine) [Moschella et al., 2023] | $0.08 \pm 0.077$ | $0.122 \pm 0.109$ | $0.21 \pm 0.149$ | $0.089 \pm 0.094$ | $0.035 \pm 0.034$ |
| RelGeo(L2) | $0.019 \pm 0.005$ | $0.046 \pm 0.016$ | $0.236 \pm 0.08$ | $0.156 \pm 0.089$ | $0.013 \pm 0.005$ |
| RelGeo(Diet) | $\mathbf{0.189} \pm 0.108$ | $\mathbf{0.241} \pm 0.117$ | $\mathbf{0.358} \pm 0.126$ | $\mathbf{0.327} \pm 0.184$ | $\mathbf{0.131} \pm 0.107$ |

`birder-project/CUB_200_2011-WDS`. Given the relatively small training set, we select 2000 points as the validation set. When reporting aggregated MRR metrics in the tables, we always exclude the diagonal entries as these are generally (close to) 1. For ImageNet-1k, we use the validation set and split it into the final train, val and test sets. Further details can be found in the provided code.

For all cases where we need to train classification heads, apart from the ones with Diet the heads are trained for 10 epochs, while the ones with Diet are trained for 50 epochs. The heads used to obtain the gometric information are trained using learning rate $5e - 4$ and batch size 64, while the heads used for stitching was trained using learning rate $1e - 4$ and batch size 32. We always use the Adam optimizer [Kingma and Ba, 2017].

When reporting stitching results, we train three classification heads and average the accuracies as the final results.

## A.4 Additional results on Vision Foundation models

We provide additional results on vision foundation models. For ablation studies, we focus on the performances of the models on CUB dataset. We refer to accuracy as *Accuracy*, symmetricized MRR based on cosine as *MRR Cosine Sym*, symmetricized MRR based on cdist as *MRR CDist Sym*, MRR based on cosine as *MRR Cosine* and MRR based on cdist as *MRR CDist*.

### A.4.1 Full results

We provide the heatmaps on the different datasets in Figure 9, Figure 10, Figure 11, Figure 12 and Figure 13.

### A.4.2 Other evaluation metrics

We provide the results of other evaluation metrics in Table 7, Table 8 and Table 9.

### A.4.3 Alternative aggregation

Here we consider an alternative way to aggregate the results, i.e. grouping by the models. The results are reported in Table 10, Table 11, Table 12, Table 13 and Table 14. In general, the observation remains: RelGeo(L2) yields good accuracies and RelGeo(Diet) yields good MRRs.

Table 9: Aggregated results of MRR CDist.

| Method | CIFAR-10 | CIFAR-100 | ImageNet-1k | CUB | SVHN |
|---|---|---|---|---|---|
| Rel(Cosine) [Moschella et al., 2023] | $0.051 \pm 0.072$ | $0.071 \pm 0.107$ | $0.078 \pm 0.105$ | $0.023 \pm 0.02$ | $0.02 \pm 0.032$ |
| RelGeo(L2) | $0.019 \pm 0.005$ | $0.04 \pm 0.015$ | $0.106 \pm 0.11$ | $0.108 \pm 0.092$ | $0.012 \pm 0.005$ |
| RelGeo(Diet) | $\mathbf{0.127} \pm 0.118$ | $\mathbf{0.151} \pm 0.138$ | $\mathbf{0.298} \pm 0.141$ | $\mathbf{0.269} \pm 0.195$ | $\mathbf{0.123} \pm 0.103$ |

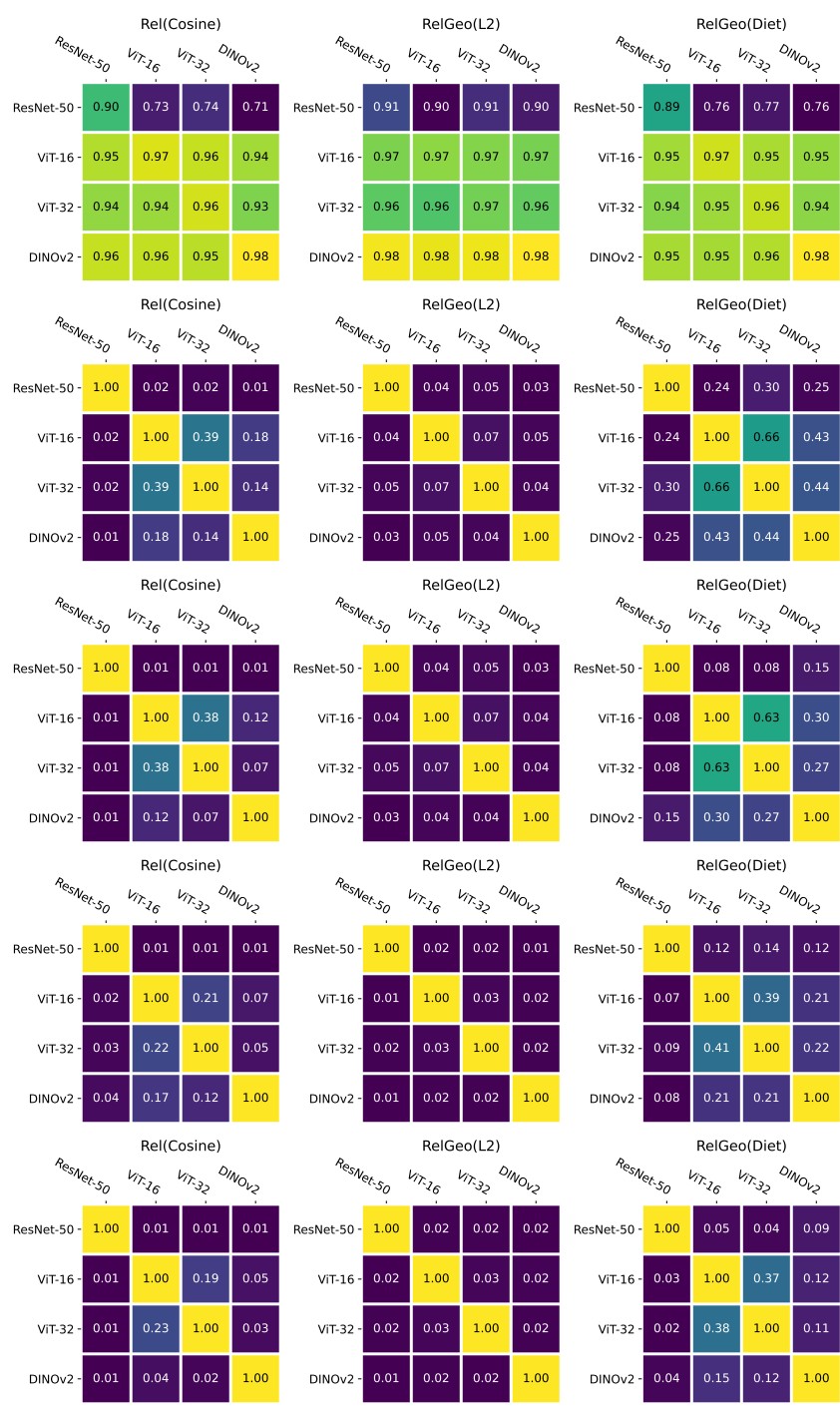

Figure 9: Results on CIFAR-10. From top to bottom: Accuracy, MRR Cosine Sym, MRR CDist Sym, MRR Cosine, MRR CDist

Table 10: Alternatively aggregated results of Accuracy.

| Method | ResNet-50 | ViT-16 | ViT-32 | DINOv2 |
|---|---|---|---|---|
| Rel(Cosine) [Moschella et al., 2023] | $0.507 \pm 0.2$ | $0.669 \pm 0.229$ | $0.664 \pm 0.218$ | $0.678 \pm 0.24$ |
| RelGeo(L2) | $\mathbf{0.646} \pm 0.209$ | $\mathbf{0.709} \pm 0.208$ | $\mathbf{0.72} \pm 0.194$ | $\mathbf{0.737} \pm 0.209$ |
| RelGeo(Diet) | $0.529 \pm 0.194$ | $0.658 \pm 0.229$ | $0.661 \pm 0.219$ | $0.668 \pm 0.237$ |

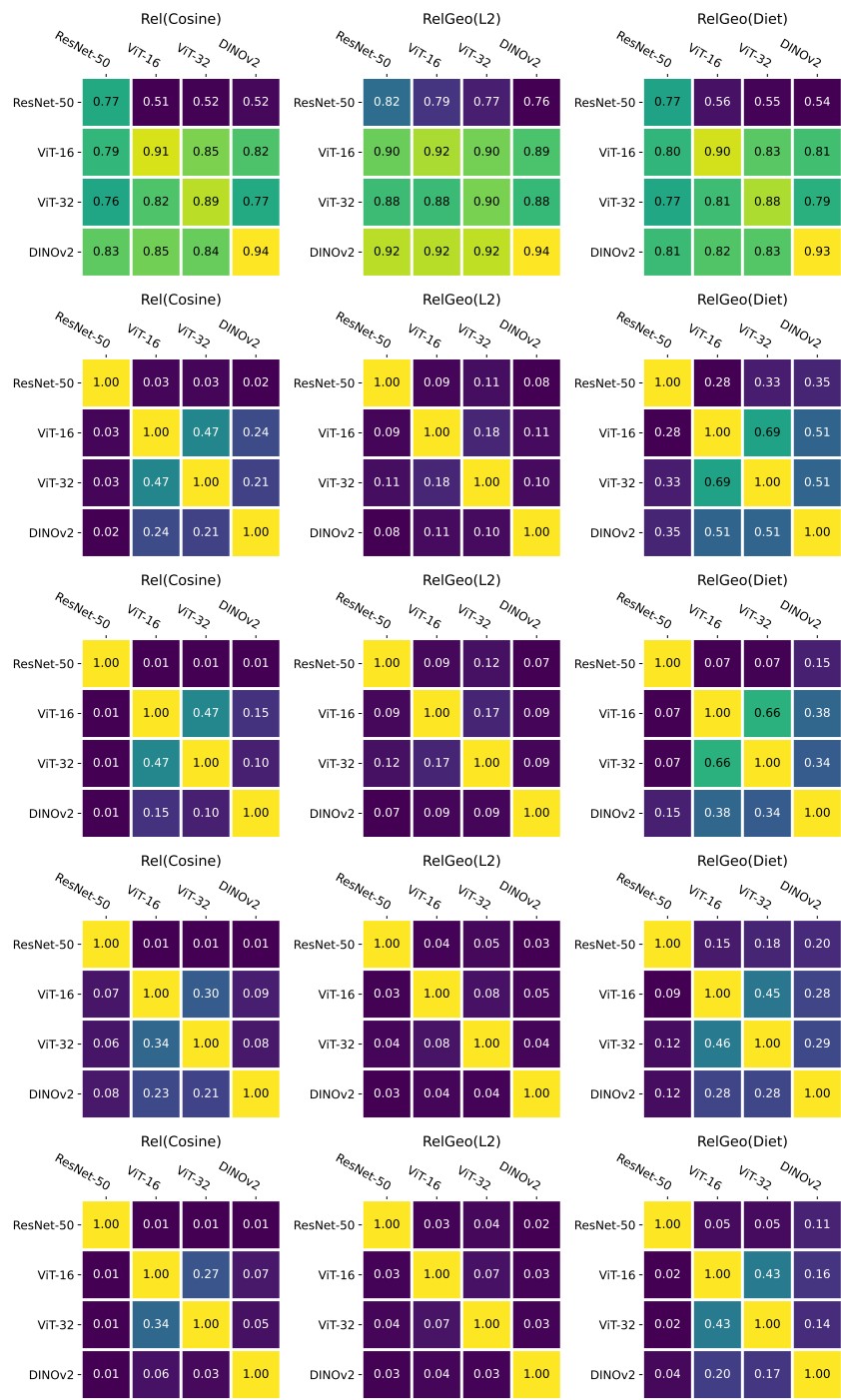

Figure 10: Results on CIFAR-100. From top to bottom: Accuracy, MRR Cosine Sym, MRR CDist Sym, MRR Cosine, MRR CDist

Table 11: Alternatively aggregated results of MRR Cosine Sym.

| Method | ResNet-50 | ViT-16 | ViT-32 | DINOv2 |
|---|---|---|---|---|
| Rel(Cosine) [Moschella et al., 2023] | $0.032 \pm 0.023$ | $0.212 \pm 0.173$ | $0.208 \pm 0.175$ | $0.124 \pm 0.104$ |
| RelGeo(L2) | $0.143 \pm 0.132$ | $0.197 \pm 0.186$ | $0.205 \pm 0.189$ | $0.154 \pm 0.137$ |
| RelGeo(Diet) | $\mathbf{0.336} \pm 0.143$ | $\mathbf{0.506} \pm 0.2$ | $\mathbf{0.526} \pm 0.187$ | $\mathbf{0.42} \pm 0.106$ |

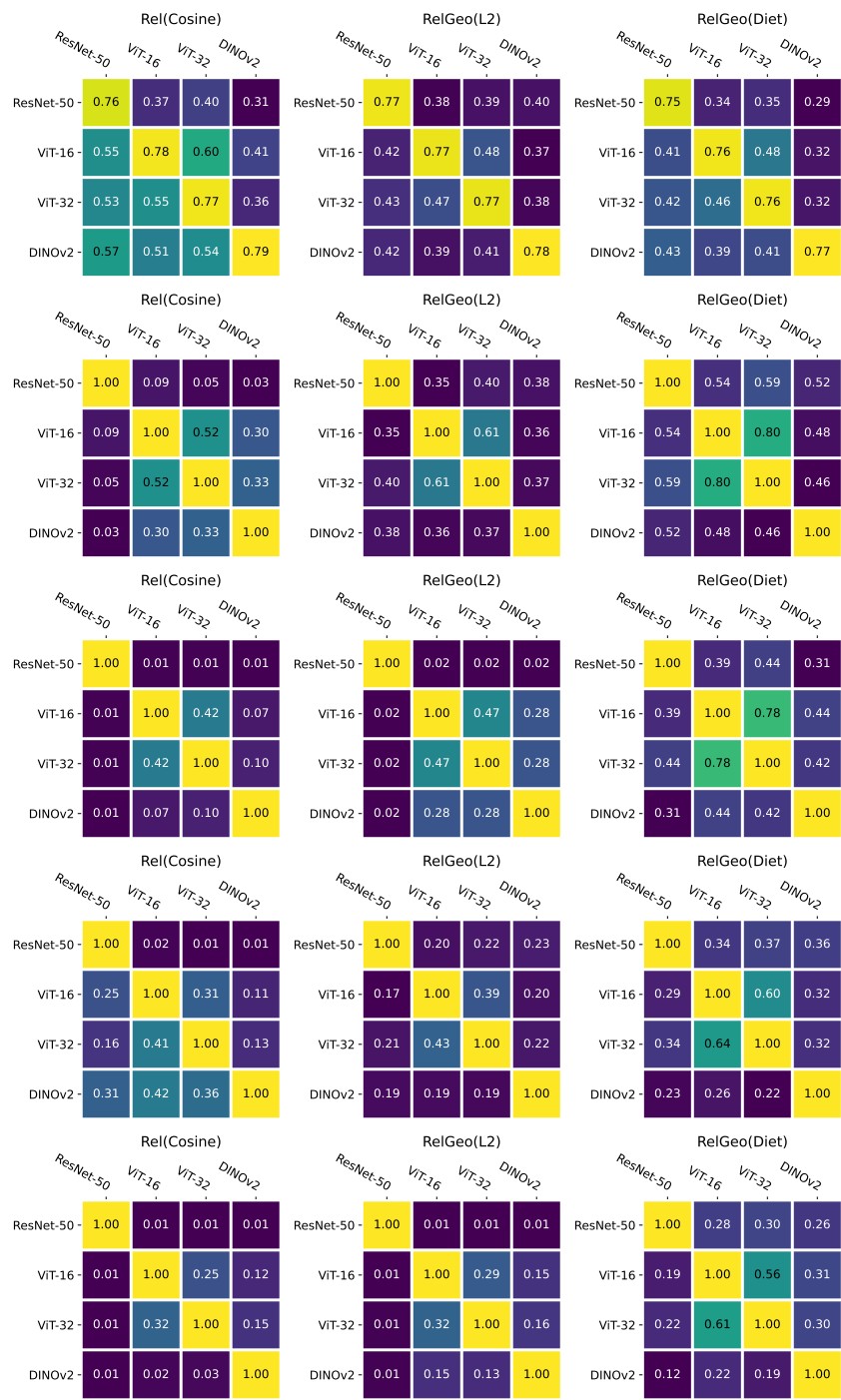

Figure 11: Results on ImageNet-1k. From top to bottom: Accuracy, MRR Cosine Sym, MRR CDist Sym, MRR Cosine, MRR CDist

Table 12: Alternatively aggregated results of MRR CDist Sym.

| Method | ResNet-50 | ViT-16 | ViT-32 | DINOv2 |
|---|---|---|---|---|
| Rel(Cosine) [Moschella et al., 2023] | $0.009 \pm 0.005$ | $0.141 \pm 0.156$ | $0.134 \pm 0.158$ | $0.049 \pm 0.047$ |
| RelGeo(L2) | $0.052 \pm 0.033$ | $0.144 \pm 0.146$ | $0.147 \pm 0.146$ | $0.103 \pm 0.085$ |
| RelGeo(Diet) | $\mathbf{0.204} \pm 0.13$ | $\mathbf{0.432} \pm 0.235$ | $\mathbf{0.437} \pm 0.232$ | $\mathbf{0.313} \pm 0.108$ |

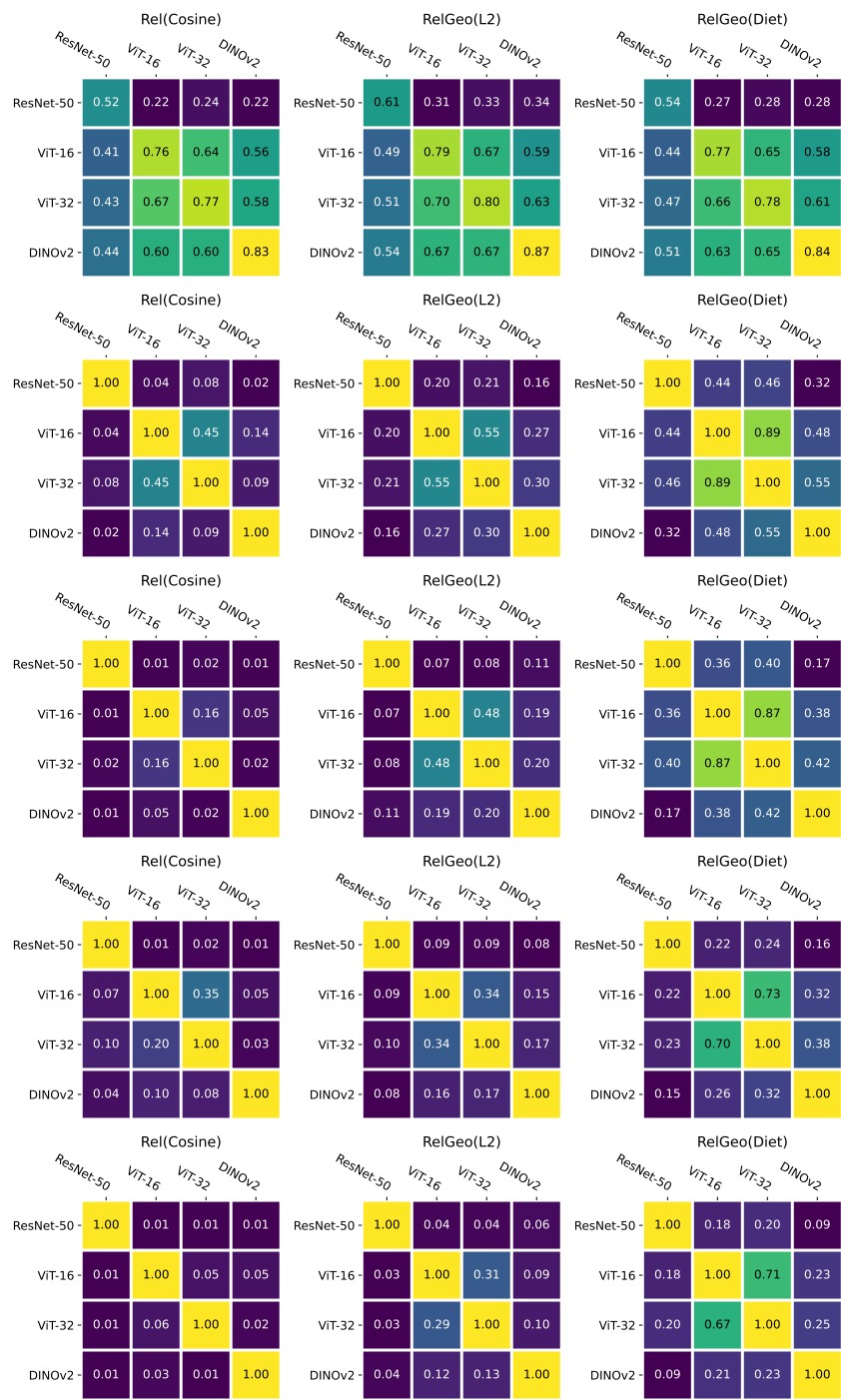

Figure 12: Results on CUB. From top to bottom: Accuracy, MRR Cosine Sym, MRR CDist Sym, MRR Cosine, MRR CDist

Table 13: Alternatively aggregated results of MRR Cosine.

| Method | ResNet-50 | ViT-16 | ViT-32 | DINOv2 |
|---|---|---|---|---|
| Rel(Cosine) [Moschella et al., 2023] | $0.011 \pm 0.005$ | $0.138 \pm 0.11$ | $0.133 \pm 0.112$ | $0.147 \pm 0.128$ |
| RelGeo(L2) | $0.074 \pm 0.077$ | $0.107 \pm 0.118$ | $0.116 \pm 0.126$ | $0.079 \pm 0.074$ |
| RelGeo(Diet) | $\mathbf{0.182} \pm 0.107$ | $\mathbf{0.299} \pm 0.184$ | $\mathbf{0.316} \pm 0.182$ | $\mathbf{0.201} \pm 0.076$ |

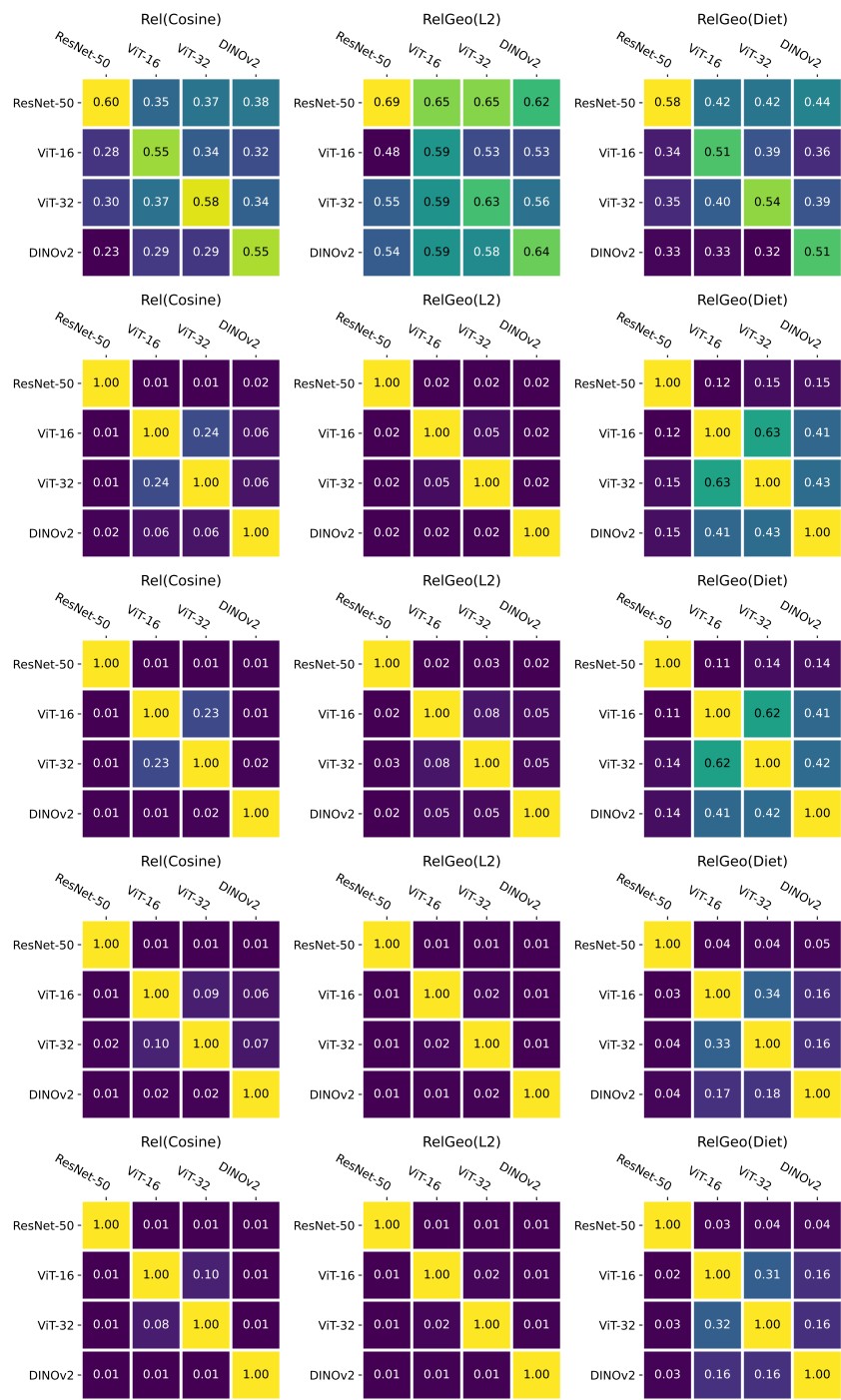

Figure 13: Results on SVHN. From top to bottom: Accuracy, MRR Cosine Sym, MRR CDist Sym, MRR Cosine, MRR CDist

Table 14: Alternatively aggregated results of MRR CDist.

| Method | ResNet-50 | ViT-16 | ViT-32 | DINOv2 |
|---|---|---|---|---|
| Rel(Cosine) [Moschella et al., 2023] | $0.007 \pm 0.001$ | $0.079 \pm 0.086$ | $0.089 \pm 0.112$ | $0.019 \pm 0.015$ |
| RelGeo(L2) | $0.023 \pm 0.016$ | $0.075 \pm 0.096$ | $0.078 \pm 0.099$ | $0.051 \pm 0.05$ |
| RelGeo(Diet) | $\mathbf{0.121} \pm 0.094$ | $\mathbf{0.253} \pm 0.193$ | $\mathbf{0.258} \pm 0.194$ | $\mathbf{0.143} \pm 0.065$ |

### A.4.4 Number of anchors

We investigate the impact of the number of anchors. The results are shown in Figure 14 and Figure 15. The general conclusion that RelGeo(L2) is good in terms of accuracies, RelGeo(Diet) is good in terms of MRRs persist with varying number of anchors.

### A.4.5 Number of Diet points

We analyze the impact of the number of Diet points. The results are shown in Figure 16. The performances of RelGeo(Diet) improve as the number of diet points become larger.

### A.4.6 Number of discretization steps

We analyze the impact of the number of discretization steps on RelGeo(L2) and RelGeo(Diet) and provide the results in Figure 17 and Figure 18. The performances do not vary much depending on the discretization steps, though using multiple steps seems to help.

### A.4.7 Diet augmentation strengths

We analyze the impact of different data augmentation strengths on RelGeo(Diet). The results are shown in Figure 19. Similar to the observations in terms of self-supervised learning [Ibrahim et al., 2024], RelGeo(Diet) benefits from stronger data augmentations.

### A.4.8 Anchor selection scheme

As discussed in Moschella et al. [2023], there are different ways of choosing the anchors. In the main paper, we consider the case where the anchors are selected uniformly at random, referred to as *uniform*. There are other choices as well, e.g. using farthest point sampling, referred to as *fps*, and using as anchors the data point close to the centroids of K-means clustering, referred to as *kmeans*.

Here we additionally report the results for *fps* and Since we need to align multiple models, in practice we use the selection mechanism to select a fixed number of anchors based on the representations of each model, and combine them while employing random subsampling to obtain the final anchors of a given number.

The experimental results for *fps* and *kmeans* are shown in Figure 20 and Figure 21, respectively.

### A.4.9 Multimodal

In the main paper we reported results based on MRR Cosine Sym. Below we show the full experimental results in Figure 22, and report aggregated results in Table 15.

Table 15: Average MRR results for different methods across different datasets. Relative representations pulling back from diet decoder (`RelGeo(Diet)`) consistently provides better retrievals.

| Method | MRR Cosine Sym | MRR CDist Sym | MRR Cosine | MRR CDist |
|---|---|---|---|---|
| `Rel(Cosine)` [Moschella et al., 2023] | $0.298 \pm 0.395$ | $0.293 \pm 0.389$ | $0.283 \pm 0.382$ | $0.252 \pm 0.373$ |
| `RelGeo(Diet)` | $\mathbf{0.413} \pm 0.353$ | $\mathbf{0.384} \pm 0.359$ | $\mathbf{0.317} \pm 0.372$ | $\mathbf{0.302} \pm 0.378$ |

### A.4.10 Running times

We report running times for autoencoder experiments with an NVIDIA 3080 Ti GPU and vision foundational model experiments with an NVIDIA A100 GPU.

**Times to train the models**    For autoencoder experiments, training the autoencoders is not expensive, as it merely takes around 20 minutes on an RTX3080Ti for 50 epochs.

For vision foundation models, the employed pretrained backbones are typically computationally costly to train. For instance, Oquab et al. [2024] reported that training DINOv2 ViT-L/14 on ImagetNet-22k using 96 A100-80GB GPUs takes approximately 3.3 days. In contrast, training the decoders is much faster. We report the running times to train the decoders in Table 16, where for each setting we

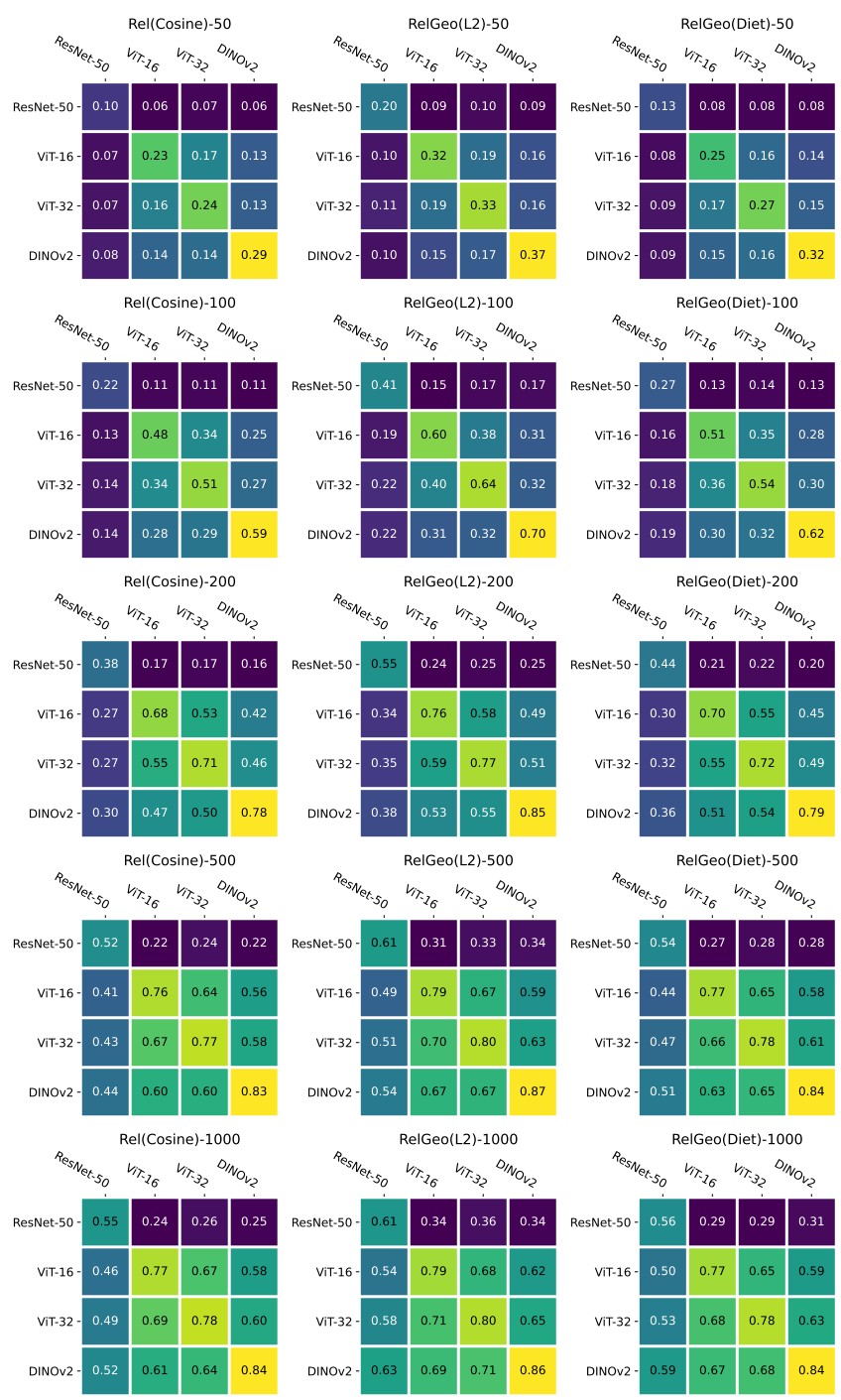

Figure 14: Accuracies on CUB with varying number of anchors. From top to bottom: 50, 100, 200, 500, 1000 anchors.

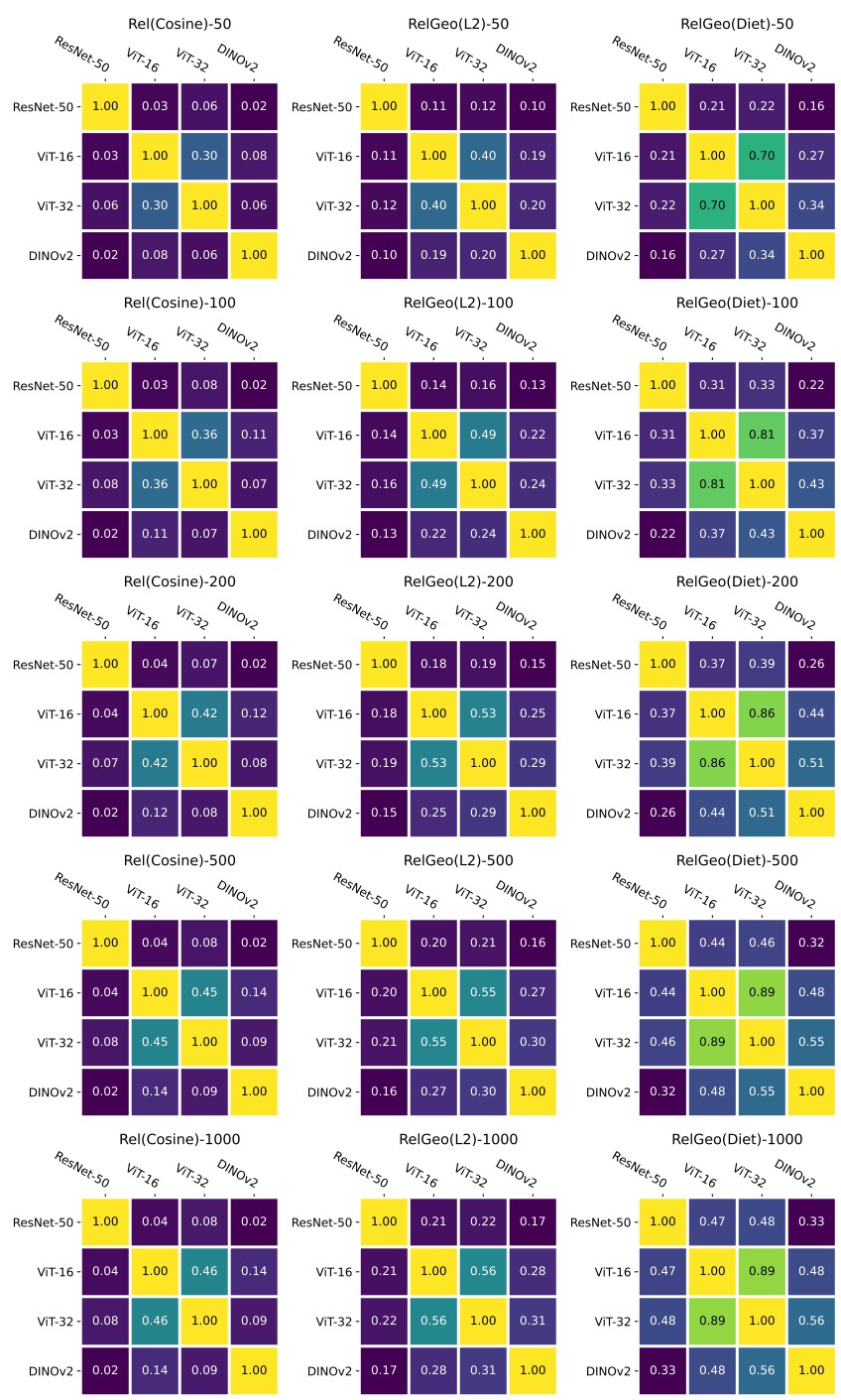

Figure 15: MRR Cosine Sym on CUB with varying number of anchors. From top to bottom: 50, 100, 200, 500, 1000 anchors.

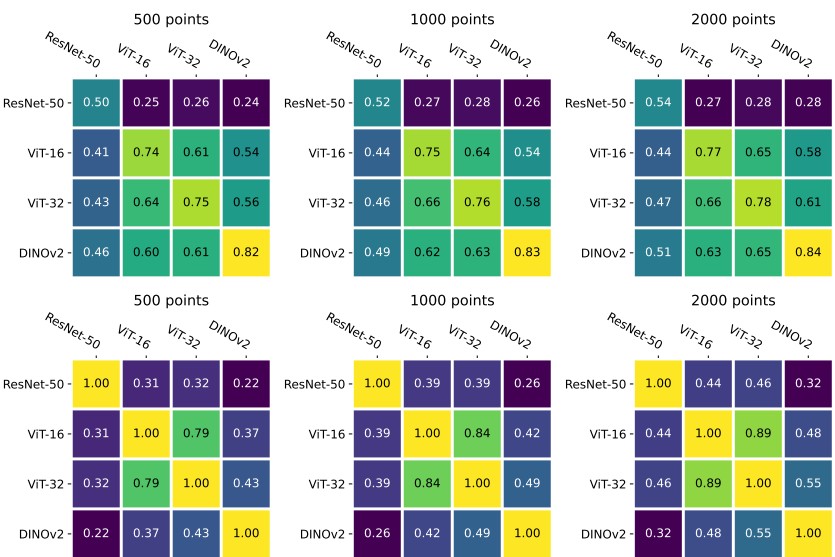

Figure 16: Results of RelGeo(Diet) on CUB with varying number of diet points. Top: accuracies; bottom: MRR Cosine Sym.

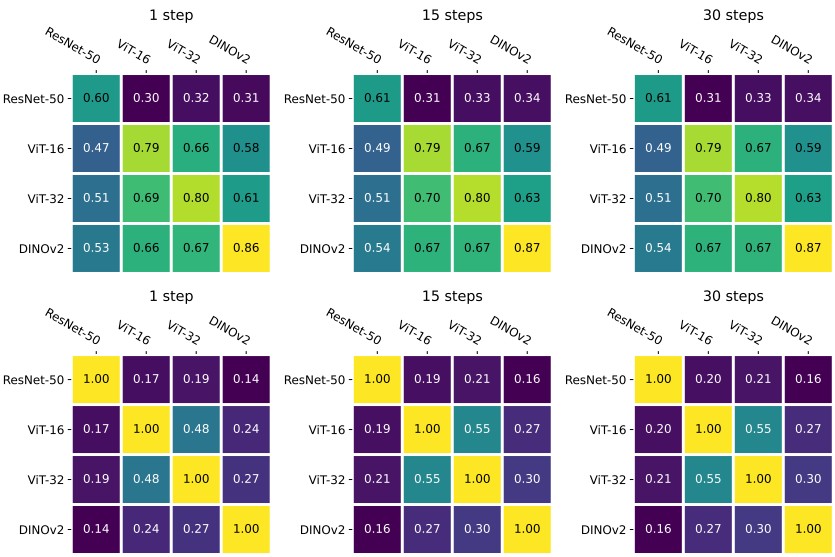

Figure 17: Results of RelGeo(L2) on CUB with varying number of discretization steps. Top: accuracies; bottom: MRR Cosine Sym.

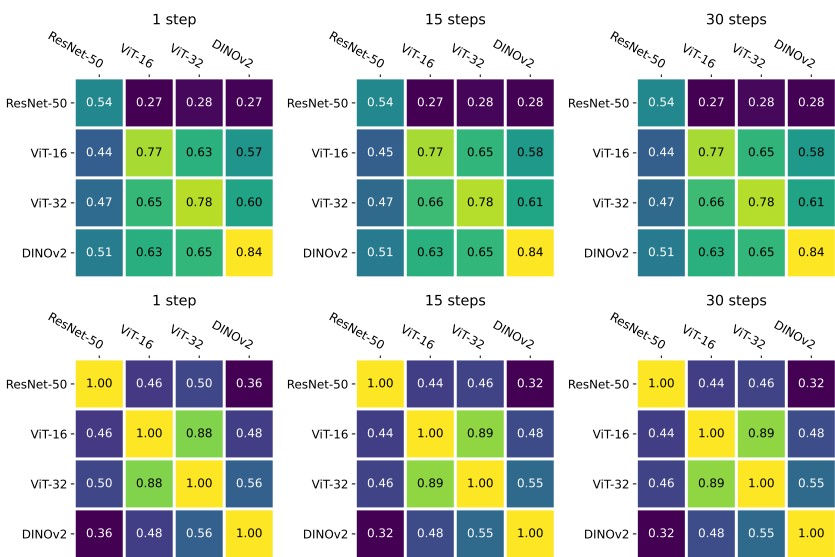

Figure 18: Results of RelGeo(Diet) on CUB with varying number of discretization steps. Top: accuracies; bottom: MRR Cosine Sym.

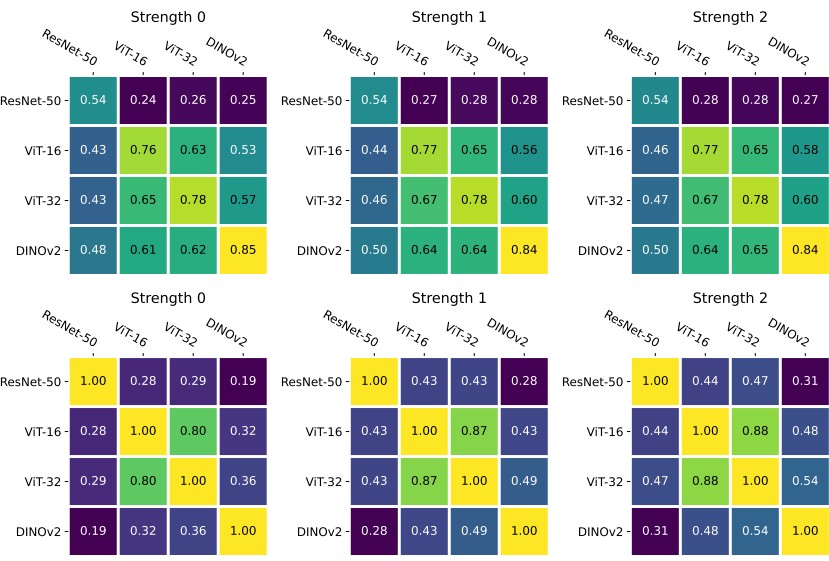

Figure 19: Results of RelGeo(Diet) on CUB with varying diet augmentation strengths. Results with strength 3 can be seen above. Top: accuracies; bottom: MRR Cosine Sym.

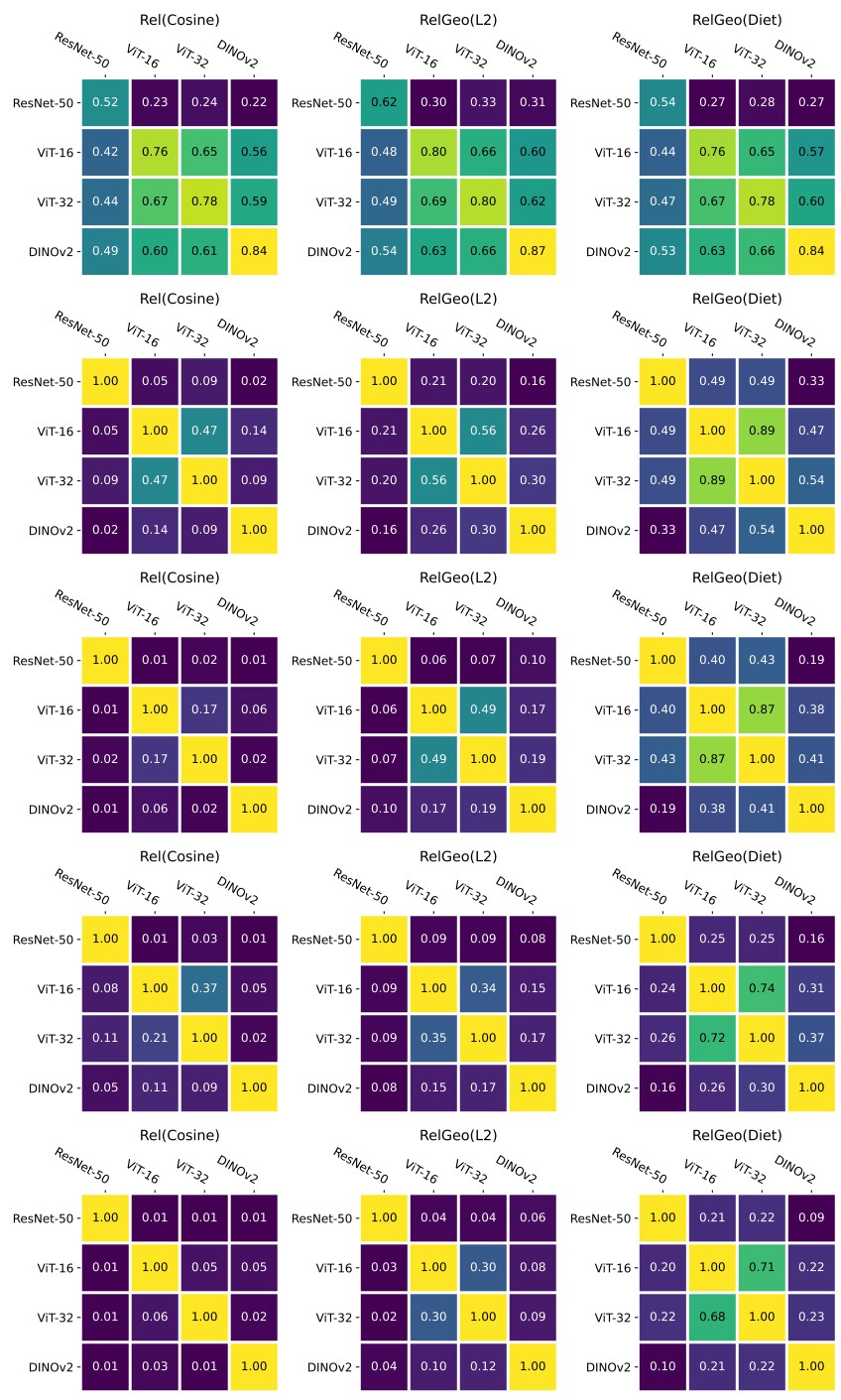

Figure 20: Results using the *fps* scheme to select the anchors. From top to bottom: accuracies, MRR Cosine Sym, MRR CDist Sym, MRR Cosine, MRR CDist.

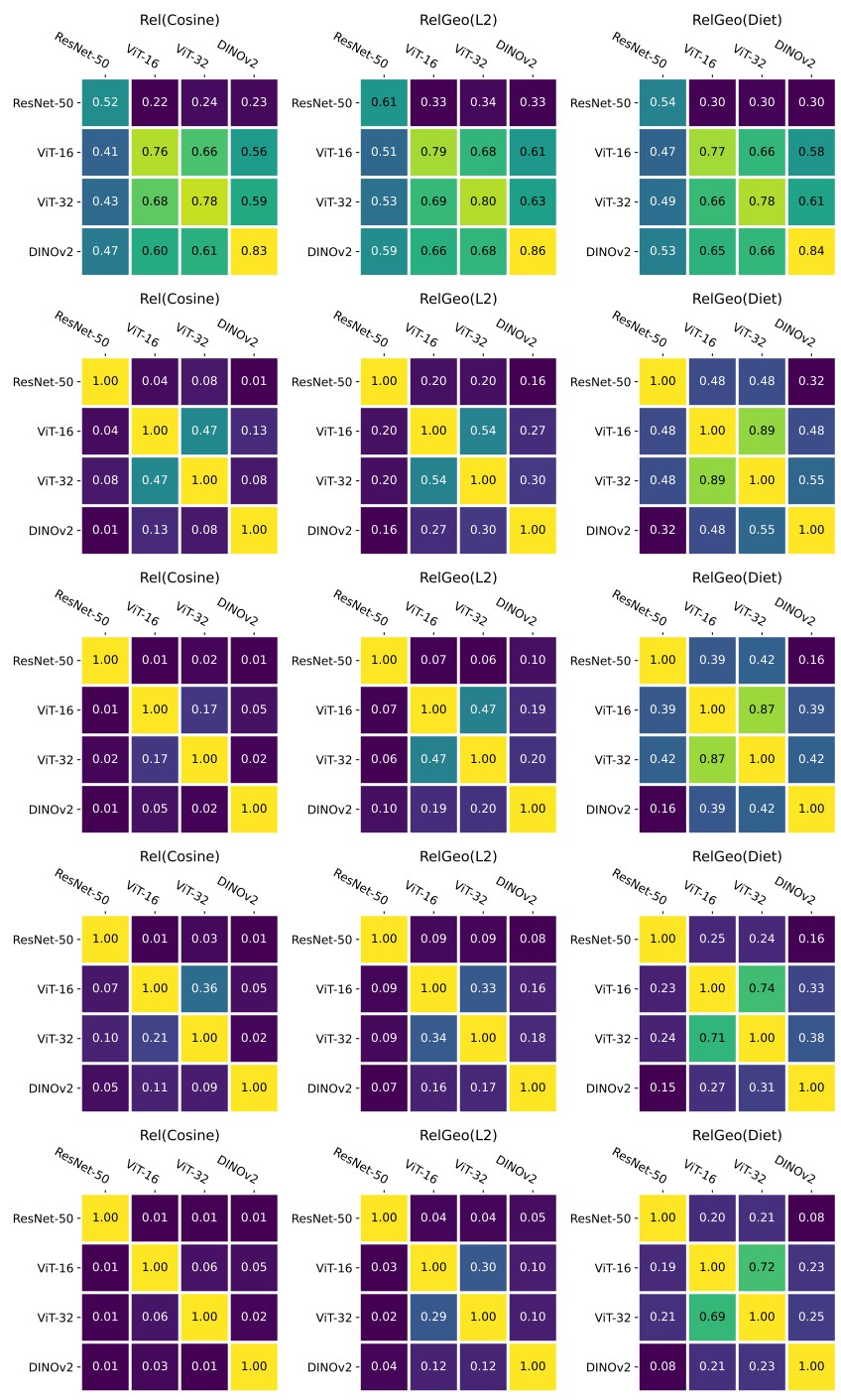

Figure 21: Results using the *kmeans* scheme to select the anchors. From top to bottom: accuracies, MRR Cosine Sym, MRR CDist Sym, MRR Cosine, MRR CDist.

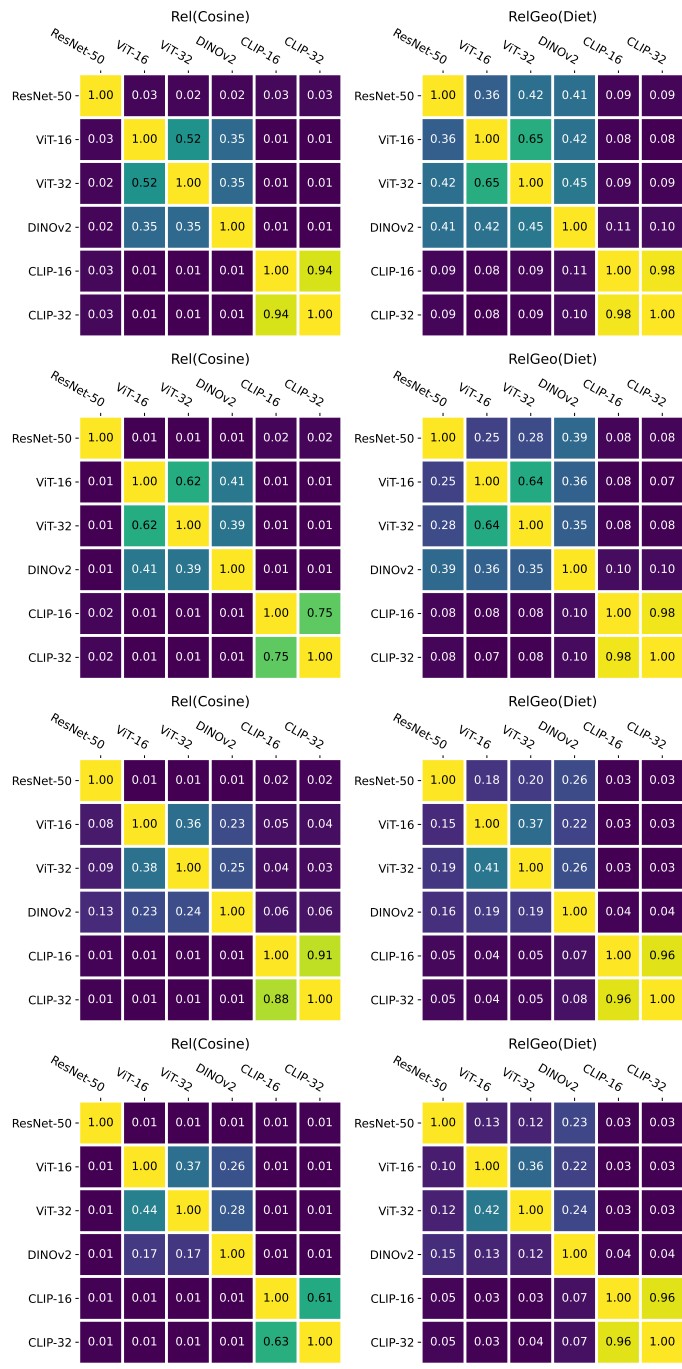

Figure 22: Stitching results for multimodal scenario. From top to bottom: MRR Cosine Sym, MRR CDist Sym, MRR Cosine, MRR CDist.

average over the different models to obtain uncertainty estimates. We remark that the exact running times depend heavily on implementation details, while currently we focus on correctness instead of the speed, and the running times could possibly be improved with better implementations.

Table 16: Times in seconds for training the decoders.

| Decoder | CIFAR-10 | CIFAR-100 | ImageNet-1k | CUB | SVHN |
|---------|----------|-----------|-------------|-----|------|
| Abs | $10.467 \pm 0.765$ | $10.71 \pm 0.655$ | $8.804 \pm 0.581$ | $1.593 \pm 0.615$ | $15.57 \pm 0.843$ |
| Diet(0) | $3.462 \pm 0.03$ | $3.501 \pm 0.045$ | $3.556 \pm 0.018$ | $3.509 \pm 0.028$ | $3.506 \pm 0.044$ |
| Diet(1) | $718.818 \pm 216.179$ | $730.446 \pm 226.91$ | $1651.49 \pm 211.037$ | $1937.656 \pm 192.894$ | $705.929 \pm 202.844$ |
| Diet(2) | $735.144 \pm 219.211$ | $739.671 \pm 222.36$ | $2034.677 \pm 276.309$ | $2428.942 \pm 196.388$ | $727.477 \pm 206.929$ |
| Diet(3) | $771.299 \pm 213.022$ | $767.453 \pm 222.62$ | $2645.914 \pm 260.069$ | $3201.588 \pm 242.339$ | $754.965 \pm 202.021$ |

**Times to obtain the representations** We first investigate the running times - accuracy tradeoff of RelGeo representations on CUB dataset, where we vary the number of anchors and monitor the times to obtain the representations and the qualities of the resulting representations. We report the results on autoencoders in Table 17 and the results on vision foundation models in Table 18, Table 19 and Table 20, respectively.

We then investigate the times to evaluate the representations across different datasets and report the results in Table 21, under the same experimental settings as reported in the main paper.

### A.4.11  RelGeo(Fisher)

We additionally report the results of relative geodesic representations based on another choice of Rimennian metric, RelGeo(Fisher), which pulls back the Fisher-Rao metric from the classification heads' output probabilities on the different datasets. We report the aggregated results in Table 22, and the results on the individual datasets in Figure 23, Figure 24, Figure 25, Figure 26 and Figure 27. RelGeo(Fisher) often results in higher accuracies and lower MRRs; we hypothesize that this is due to the Neural Collapse phenomenon [Kothapalli, 2023] observed in well-trained neural networks.

Table 17: Running time / accuracy tradeoff of RelGeo(L2).

| Num Anchors | Time (s) $\pm$ Std | MRR |
|-------------|--------------------|-----|
| 2 | $0.8480 \pm 0.0407$ | 0.0168 |
| 3 | $0.8348 \pm 0.0367$ | 0.0807 |
| 5 | $0.8377 \pm 0.0435$ | 0.3503 |
| 8 | $0.9450 \pm 0.0285$ | 0.7004 |
| 10 | $1.0969 \pm 0.0297$ | 0.8384 |
| 15 | $1.2853 \pm 0.0264$ | 0.9296 |
| 20 | $1.6160 \pm 0.0188$ | 0.9616 |
| 25 | $1.8548 \pm 0.0218$ | 0.9868 |
| 50 | $3.3311 \pm 0.0286$ | 0.9981 |
| 100 | $6.2122 \pm 0.0318$ | 0.9986 |
| 300 | $17.6494 \pm 0.0653$ | 0.9982 |
| 500 | $29.1925 \pm 0.0806$ | 0.9986 |

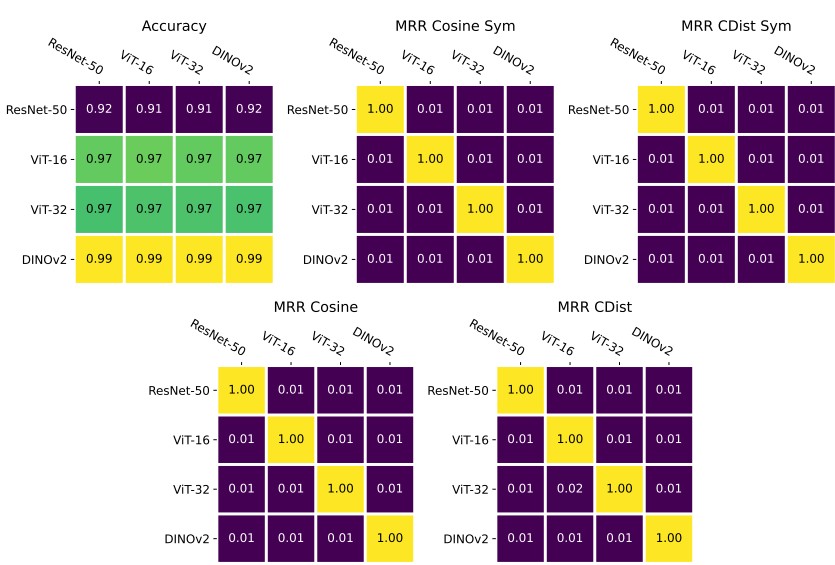

Figure 23: Results of RelGeo(Fisher) on CIFAR-10.

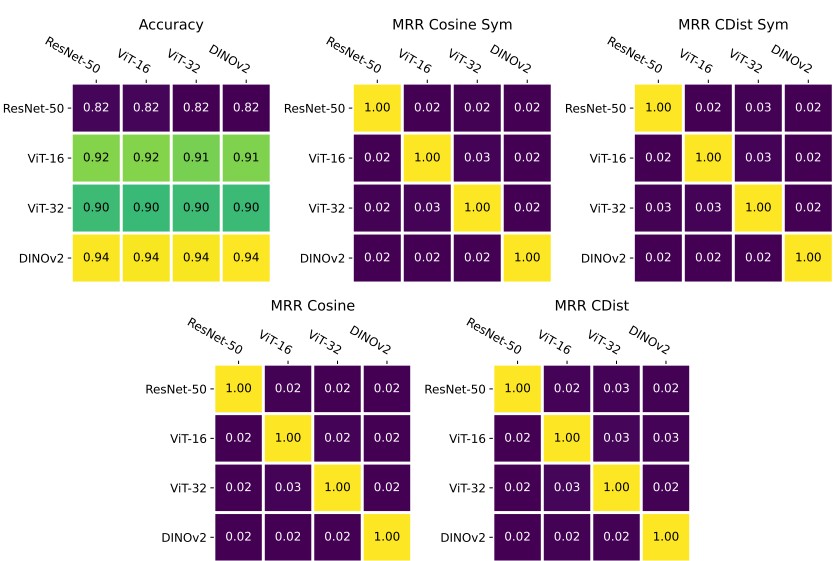

Figure 24: Results of RelGeo(Fisher) on CIFAR-100.

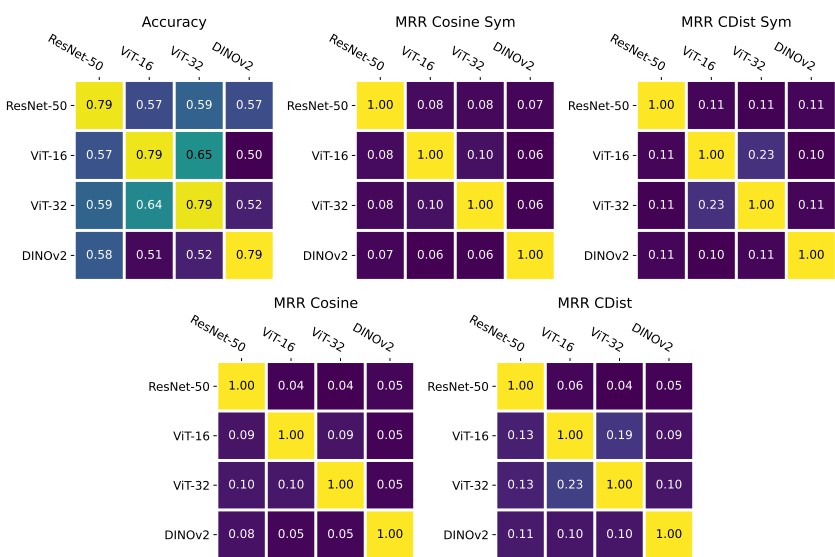

Figure 25: Results of RelGeo(Fisher) on ImageNet-1k.

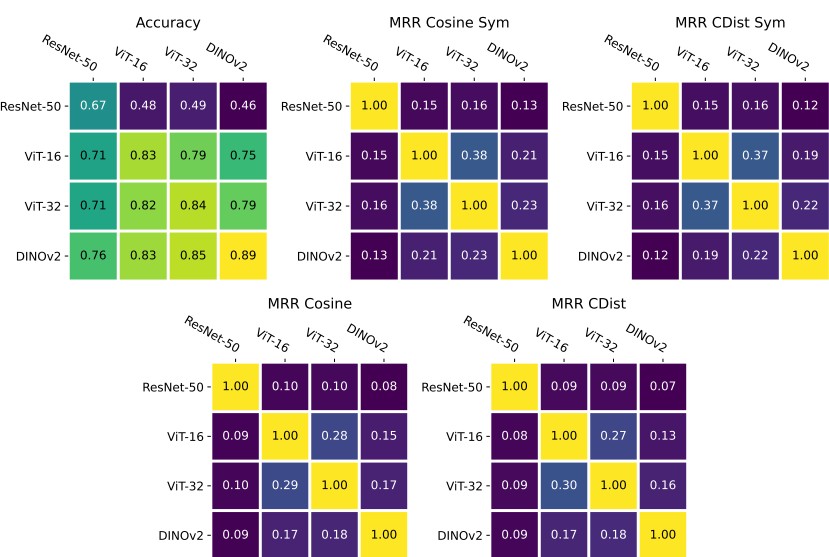

Figure 26: Results of RelGeo(Fisher) on CUB.

Table 18: Running time / accuracy tradeoff of Rel(Cosine).

| Num Anchors | Time (s) | Accuracy |
|---|---|---|
| 200 | $0.088 \pm 0.14$ | $0.425 \pm 0.189$ |
| 500 | $0.175 \pm 0.053$ | $0.531 \pm 0.188$ |
| 1000 | $0.062 \pm 0.091$ | $0.559 \pm 0.179$ |

Table 19: Running time / accuracy tradeoff of RelGeo(L2).

| Num Anchors | Time (s) | Accuracy |
|---|---|---|
| 200 | $8.004 \pm 3.304$ | $0.5 \pm 0.184$ |
| 500 | $21.297 \pm 8.774$ | $0.595 \pm 0.163$ |
| 1000 | $47.218 \pm 19.486$ | $0.619 \pm 0.154$ |

Table 20: Running time / accuracy tradeoff of RelGeo(Diet).

| Num Anchors | Time (s) | Accuracy |
|---|---|---|
| 200 | $7.274 \pm 3.28$ | $0.459 \pm 0.177$ |
| 500 | $19.427 \pm 8.771$ | $0.559 \pm 0.171$ |
| 1000 | $43.108 \pm 19.502$ | $0.585 \pm 0.163$ |

Table 21: Times in seconds of `Rel(Cosine)`, `RelGeo(L2)` and `RelGeo(Diet)` for generating the representations.

| Method | CIFAR-10 | CIFAR-100 | ImageNet-1k | CUB | SVHN |
|---|---|---|---|---|---|
| `Rel(Cosine)` [Moschella et al., 2023] | $0.071 \pm 0.113$ | $0.184 \pm 0.065$ | $0.085 \pm 0.124$ | $0.05 \pm 0.071$ | |
| `RelGeo(L2)` | $85.414 \pm 40.61$ | $86.079 \pm 40.652$ | $258.788 \pm 70.064$ | $139.998 \pm 66.509$ | |
| `RelGeo(Diet)` | $89.899 \pm 40.599$ | $89.902 \pm 40.588$ | $155.991 \pm 70.136$ | $147.336 \pm 66.505$ | |

Table 22: Results of RelGeo(Fisher).

| Metric | CIFAR-10 | CIFAR-100 | ImageNet-1k | CUB | SVHN |
|---|---|---|---|---|---|
| Accuracy | $0.959 \pm 0.026$ | $0.894 \pm 0.043$ | $0.623 \pm 0.103$ | $0.729 \pm 0.133$ | $0.625 \pm 0.033$ |
| MRR Cosine Sym | $0.012 \pm 0.0$ | $0.021 \pm 0.003$ | $0.075 \pm 0.014$ | $0.211 \pm 0.082$ | $0.034 \pm 0.01$ |
| MRR CDist Sym | $0.012 \pm 0.001$ | $0.025 \pm 0.004$ | $0.13 \pm 0.046$ | $0.201 \pm 0.08$ | $0.028 \pm 0.008$ |
| MRR Cosine | $0.012 \pm 0.001$ | $0.019 \pm 0.003$ | $0.066 \pm 0.022$ | $0.152 \pm 0.069$ | $0.011 \pm 0.002$ |
| MRR CDist | $0.012 \pm 0.001$ | $0.023 \pm 0.005$ | $0.111 \pm 0.052$ | $0.144 \pm 0.073$ | $0.011 \pm 0.003$ |

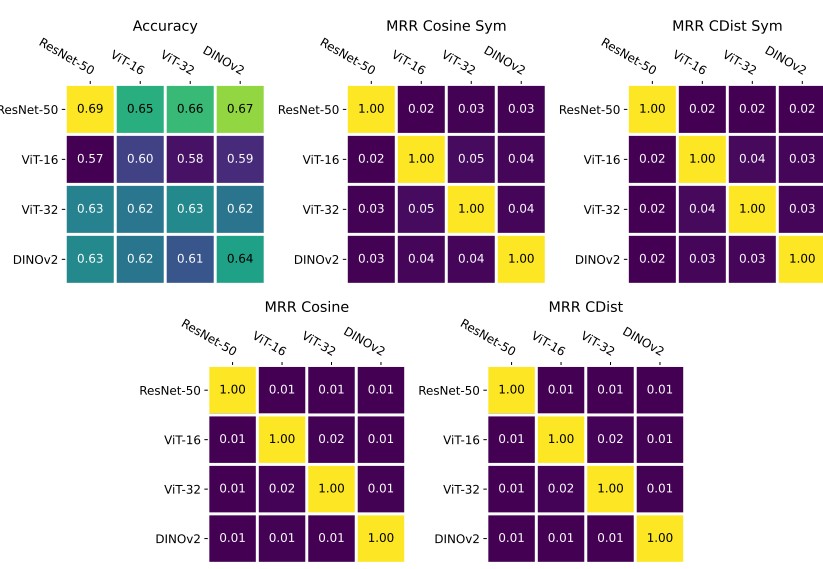

Figure 27: Results of RelGeo(Fisher) on SVHN.

