# OpenReview forum: "Connecting Neural Models Latent Geometries with Relative Geodesic Representations"
_NeurIPS.cc/2025/Conference — NeurIPS 2025 poster_

### Official Review · Reviewer_k3Ho · 2025-06-23

**Clarity:** 2
**Significance:** 3
**Originality:** 3
**Rating:** 3
**Confidence:** 3

**Summary:**

The paper introduces Relative Geodesic Representations, a framework for aligning the latent spaces of independently trained neural networks. The authors endow each latent space with a Riemannian metric obtained by pulling back a metric defined in output space. Then, they compute approximate geodesic lengths along straight-line latent paths to construct distance-based relative feature vectors for a small set of anchor points. They demonstrate that these representations are approximately invariant to isometric transformations that link the two latent spaces. They tackle both the retrieval and identification of corresponding points, as well as zero-shot module stitching across models, encompassing auto-encoders, ResNet/ViT classifiers, and DINOv2 self-supervised encoders. Across five vision datasets, the proposed method, especially the variant that pulls back a spherical metric learned with the Diet self-supervised head, consistently improves mean-reciprocal-rank retrieval and stitched accuracy over prior cosine-based relative representations, while requiring only a handful of anchors.

**Questions:**

1. Experiments use random anchors; have you tried diversity-based or class-balanced sampling, and how does performance vary with anchor quality? Demonstrating robustness or providing guidance could improve clarity.
2. Your approximation relies on a fixed number of steps N (Eq. 4–5). How sensitive are alignment and stitching metrics to N, and is there an adaptive criterion that balances accuracy and efficiency?
3. You suggest that relative geodesics could aid multimodal alignment. What modifications (if any) are needed to handle heterogeneous output geometries, aligning a vision encoder with a text encoder, and do preliminary results support this claim?
4. Proposition 3.1 guarantees length invariance under re-parametrisation, but the paper does not discuss how many anchors are theoretically sufficient. Is it possible to bound the number of anchors needed so that the relative geodesic embedding preserves all pairwise geodesic ranks with a specific probability?
5. The method is invariant to isometries of the underlying manifold. Which non-isometric transformations (e.g., local dilations, shears) break the relative geodesic representation, and can you characterize their practical prevalence in modern networks?

**Ethical Concerns:**

["NO or VERY MINOR ethics concerns only"]

**Final Justification:**

I raise the score on originality from 2 to 3.

**Limitations:**

The paper includes a “Limitations and future work” subsection, and the checklist discusses broader impacts.

**Paper Formatting Concerns:**

No major formatting issue.

**Quality:**

3

**Strengths And Weaknesses:**

Strengths:
1. Quality: Solid formulation grounded in differential geometry; extensive experiments on both small and large-scale foundation models with multiple backbones; competitive baselines; standard deviations reported.
2. Clarity: Writing and contributions are generally clear, with well-designed figures.
3. Significance: Bridges latent-space geometry and cross-model alignment, yielding a unifying perspective. Has potential practical payoff for model reuse without training data.
4. Originality: Extends Relative Representations by replacing cosine with intrinsic geodesic distances and proposing a scalable proxy, the first to demonstrate that pull-back metrics from discriminative/self-supervised heads improve cross-model tasks.

Weakness:
1. Quality: No quantitative error analysis; computational cost (seconds / GPU-hours) is not included.
2. Clarity: Dense notation (multiple pull-back metrics) may overwhelm non-geometric readers.
3. Significance: The impact is currently demonstrated only for vision; the benefit for language or multimodal models remains speculative.
4. Originality: Builds directly on prior relative-representation framework; novelty is incremental in that sense.

---

> ### Author Rebuttal · Authors · 2025-07-31
>
> We thank the reviewers for their feedback and suggestions. Below we address the Reviewer concerns. We remain available for any further clarification during the discussion period.
>
> **Anchors sampling strategies**: We report the accuracies and symmetricized MRR based on Cosine on CUB dataset, comparing uniform sampling for anchors (as used in the main paper) with using farthest point sampling (fps) and kmeans, as described in Moschella et al 2023. We observe that the general conclusion persists. Below we report the stitching accuracies; the other results based on retrieval behaved similarly, and have not been included for space constraints:  we included the full results in the manuscript.
>
> ### Accuracies
> #### Rel(Cosine)
> ##### uniform
> |           | ResNet-50 | VIT-16  | VIT-32  | DINOv2  |
> | --------- | --------- | ------- | ------- | ------- |
> | ResNet-50 | $0.519$   | $0.216$ | $0.241$ | $0.218$ |
> | VIT-16    | $0.408$   | $0.760$ | $0.645$ | $0.559$ |
> | VIT-32    | $0.426$   | $0.671$ | $0.773$ | $0.581$ |
> | DINOv2    | $0.443$   | $0.602$ | $0.605$ | $0.833$ |
>
> ##### fps
> |           | ResNet-50 | VIT-16  | VIT-32  | DINOv2  |
> | --------- | --------- | ------- | ------- | ------- |
> | ResNet-50 | $0.523$   | $0.230$ | $0.242$ | $0.219$ |
> | VIT-16    | $0.419$   | $0.764$ | $0.653$ | $0.563$ |
> | VIT-32    | $0.438$   | $0.671$ | $0.782$ | $0.592$ |
> | DINOv2    | $0.489$   | $0.597$ | $0.611$ | $0.838$ |
>
> ##### kmeans
> |           | ResNet-50 | VIT-16  | VIT-32  | DINOv2  |
> | --------- | --------- | ------- | ------- | ------- |
> | ResNet-50 | $0.522$   | $0.220$ | $0.237$ | $0.234$ |
> | VIT-16    | $0.412$   | $0.764$ | $0.657$ | $0.560$ |
> | VIT-32    | $0.430$   | $0.682$ | $0.780$ | $0.592$ |
> | DINOv2    | $0.475$   | $0.597$ | $0.611$ | $0.834$ |
>
>
> #### RelGeo(Pullback)
> ##### uniform
> |           | ResNet-50 | VIT-16  | VIT-32  | DINOv2  |
> | --------- | --------- | ------- | ------- | ------- |
> | ResNet-50 | $0.614$   | $0.312$ | $0.332$ | $0.335$ |
> | VIT-16    | $0.487$   | $0.793$ | $0.667$ | $0.593$ |
> | VIT-32    | $0.515$   | $0.695$ | $0.804$ | $0.626$ |
> | DINOv2    | $0.538$   | $0.670$ | $0.673$ | $0.866$ |
>
> ##### fps
> |           | ResNet-50 | VIT-16  | VIT-32  | DINOv2  |
> | --------- | --------- | ------- | ------- | ------- |
> | ResNet-50 | $0.615$   | $0.299$ | $0.327$ | $0.312$ |
> | VIT-16    | $0.476$   | $0.796$ | $0.656$ | $0.599$ |
> | VIT-32    | $0.491$   | $0.690$ | $0.805$ | $0.617$ |
> | DINOv2    | $0.543$   | $0.634$ | $0.659$ | $0.868$ |
>
> ##### kmeans
> |           | ResNet-50 | VIT-16  | VIT-32  | DINOv2  |
> | --------- | --------- | ------- | ------- | ------- |
> | ResNet-50 | $0.609$   | $0.329$ | $0.341$ | $0.332$ |
> | VIT-16    | $0.511$   | $0.793$ | $0.682$ | $0.614$ |
> | VIT-32    | $0.530$   | $0.689$ | $0.802$ | $0.631$ |
> | DINOv2    | $0.591$   | $0.665$ | $0.682$ | $0.865$ |
>
>
> #### RelGeo(Diet)
> ##### uniform
> |           | ResNet-50 | VIT-16  | VIT-32  | DINOv2  |
> | --------- | --------- | ------- | ------- | ------- |
> | ResNet-50 | $0.539$   | $0.242$ | $0.255$ | $0.249$ |
> | VIT-16    | $0.429$   | $0.763$ | $0.630$ | $0.533$ |
> | VIT-32    | $0.432$   | $0.653$ | $0.780$ | $0.567$ |
> | DINOv2    | $0.483$   | $0.610$ | $0.619$ | $0.847$ |
>
> ##### fps
> |           | ResNet-50 | VIT-16  | VIT-32  | DINOv2  |
> | --------- | --------- | ------- | ------- | ------- |
> | ResNet-50 | $0.544$   | $0.248$ | $0.260$ | $0.244$ |
> | VIT-16    | $0.407$   | $0.764$ | $0.623$ | $0.530$ |
> | VIT-32    | $0.445$   | $0.645$ | $0.782$ | $0.578$ |
> | DINOv2    | $0.501$   | $0.611$ | $0.632$ | $0.849$ |
>
> ##### kmeans
> |           | ResNet-50 | VIT-16  | VIT-32  | DINOv2  |
> | --------- | --------- | ------- | ------- | ------- |
> | ResNet-50 | $0.540$   | $0.269$ | $0.281$ | $0.269$ |
> | VIT-16    | $0.435$   | $0.762$ | $0.640$ | $0.548$ |
> | VIT-32    | $0.456$   | $0.646$ | $0.775$ | $0.566$ |
> | DINOv2    | $0.511$   | $0.612$ | $0.628$ | $0.849$ |
>
>
> **“Sensitive .. to N”**: we provided an ablation study of the effect of N in Appendix A.2.6. We observe that the proposed method is not too sensitive to N.
>
> **“Multimodal”**: we report the results of Rel(Cosine) and RelGeo(Diet) on aligning vision models and the text encoder of CLIP on Flickr30k. We use 2000 Diet data points, and simply use the first text string of each data point for the text part. We report the retrieval results on 10000 test data points. As shown in the result tables, RelGeo(Diet) outperforms Rel(Cosine) also in the multimodal setting. We will incorporate the full results of the experiment in the manuscript.
>
> ### Symmetricized MRR Cosine
> #### Cos
> |           | ResNet-50 | VIT-16  | VIT-32  | DINOv2  | CLIP-16 | CLIP-32 |
> | --------- | --------- | ------- | ------- | ------- | ------- | ------- |
> | ResNet-50 | $1.000$   | $0.026$ | $0.024$ | $0.025$ | $0.032$ | $0.032$ |
> | VIT-16    | $0.026$   | $1.000$ | $0.515$ | $0.347$ | $0.011$ | $0.010$ |
> | VIT-32    | $0.024$   | $0.515$ | $1.000$ | $0.350$ | $0.012$ | $0.011$ |
> | DINOv2    | $0.025$   | $0.347$ | $0.350$ | $1.000$ | $0.013$ | $0.013$ |
> | CLIP-16   | $0.032$   | $0.011$ | $0.012$ | $0.013$ | $1.000$ | $0.942$ |
> | CLIP-32   | $0.032$   | $0.010$ | $0.011$ | $0.013$ | $0.942$ | $1.000$ |
>
> #### RelGeo(Diet)
> |           | ResNet-50 | VIT-16  | VIT-32  | DINOv2  | CLIP-16 | CLIP-32 |
> | --------- | --------- | ------- | ------- | ------- | ------- | ------- |
> | ResNet-50 | $1.000$   | $0.356$ | $0.415$ | $0.415$ | $0.093$ | $0.089$ |
> | VIT-16    | $0.356$   | $1.000$ | $0.652$ | $0.425$ | $0.085$ | $0.077$ |
> | VIT-32    | $0.415$   | $0.652$ | $1.000$ | $0.446$ | $0.092$ | $0.088$ |
> | DINOv2    | $0.415$   | $0.425$ | $0.446$ | $1.000$ | $0.108$ | $0.104$ |
> | CLIP-16   | $0.093$   | $0.085$ | $0.092$ | $0.108$ | $1.000$ | $0.982$ |
> | CLIP-32   | $0.089$   | $0.077$ | $0.088$ | $0.104$ | $0.982$ | $1.000$ |
>
> **“Proposition 3.1”**: Proposition 3.1 states that each geodesic distance as used by relative geodesic representations is invariant under re-parameterisations, which holds regardless of the number of anchors.
>
> **“Non-isometric transformation”**: the Riemannian geometry framework only provides invariance under isometries. As such, it is unclear what happens under non-isometric transformations. We remark that, suppose we view each neural network as a smooth function, up to some assumptions, invariance under isometries can provide certain identifiable guarantees; see e.g. [a].
>
> _[a] Identifying metric structures of deep latent variable models, Syrota et al., ICML 2025_

---

> > ### Comment · Reviewer_k3Ho · 2025-08-02
> >
> > I want to thank the authors for their responses and the extended empirical results. The authors have addressed most of my problems and I will reconsider the scores.

---

> > > ### Author Response · Authors · 2025-08-04
> > > **Response to Reviewer's comment**
> > >
> > > We thank the Reviewer for their engagement, positive comments and for reconsidering their score.

---

### Official Review · Reviewer_age6 · 2025-07-02

**Clarity:** 2
**Significance:** 3
**Originality:** 3
**Rating:** 5
**Confidence:** 4

**Summary:**

The paper aims at problems like model stitching or retrieval, in which latent distributions or latent vectors learned by one model have to be compared/stitched to the latent space of another model. The proposal of the paper is conceptually straightforward, it is to take inspiration from Riemannian geometry and to use pullback metrics on the data space, which would work if the models under consideration in stitching are learning the same underlying manifold or distribution. This assumption in practice is just a wishful thinking approximation, however the method inspired by it can still work nevertheless. The paper shows that in fact this is the case, and the above basic idea using approximate geodesic distance has good performance.

**Questions:**

1) How fast is the computation of formulas (4) and (5) from the paper? And what value of N do you choose in the experiments? Why do you use that choices?

2) Line 199-200: you use the sphere distance for Diet, and you say later (line 205-207) that this is not fully justified ("is a strong assumption") but you say in line 206-207 that this "results in meaningful representations" where meaningful means what?. Can you point to what this means explicitly/quantitatively in what part of the paper?

**Ethical Concerns:**

["NO or VERY MINOR ethics concerns only"]

**Final Justification:**

Following the discussion with the authors, I've become more confident (confidence from 2 to 4) on the paper and I think that I can safely raise my score from 4 to 5.

**Limitations:**

I think this part does not apply to this paper.

**Paper Formatting Concerns:**

no concerns

**Quality:**

3

**Strengths And Weaknesses:**

The strength of the paper is that the experiments work, therefore the somehow bold hypothesis that a same underlying manifold characterizes to a strong degree same data seen by different models, is robust enough that the differential geometry insights on which the paper's methods are based resist and give good accuracy.

The weakness for me is that we don't quite have a clean explanation of why that works. It is known that the manifold hypothesis is not quite true in general, so some explanation seems to be wanting.

---

> ### Author Rebuttal · Authors · 2025-07-31
>
> We thank the reviewer for their feedback and suggestions. We address their questions in the following, and remain available for any further clarification in the discussion period.
>
> “**Why it works**”: Prior work on representational alignment has shown that representations from different models can often be approximately aligned using simple transformations (e.g., linear, orthogonal, or locally linear maps). Even when models are trained independently—with different architectures, modalities, or datasets that nonetheless share an underlying structure—they tend to learn similar representations, suggesting convergence toward a shared encoding of entities (Huh et al., 2024).
> From a theoretical standpoint, identifiability results [a] imply that if two discriminative models learn the same likelihood function, their internal representations must be equivalent up to a linear transformation. However, this ideal scenario rarely holds exactly in practice: training dynamics, nuisance factors, and unmodeled variability can all introduce distortions.
> In our case, it may be too strong to assume that two models learn different parametrizations of an identical manifold. Instead, we adopt a weaker assumption—that they do so up to some bounded distortion. Recent theoretical work has begun to explore relaxations of strict identifiability to account for such bounded distortions [b]. Integrating these relaxations into our Riemannian framework presents a promising direction for future work.
>
> *[a] Roeder, Geoffrey, Luke Metz, and Durk Kingma. "On linear identifiability of learned representations." International Conference on Machine Learning. PMLR, 2021.*
>
> *[b] Nielsen, Beatrix MG, et al. "When Does Closeness in Distribution Imply Representational Similarity? An Identifiability Perspective." arXiv preprint arXiv:2506.03784 (2025).*
>
> **How fast is the computation of formulas (4) and (5)**: the computation of the relative geodesic representation for a single sample requires $N \times | \mathcal{A}_X |$  forward passes of the decoder, where $N$ is the number of steps, and $| \mathcal{A}_X |$ is the number of anchors. Notice that these are vectorizable and can be run all in parallel on GPU, allowing for fast evaluation.  The number of steps we employ in the experiments is 15. In Figure 15, 16 in the Appendix, we provide an ablation on N on stitching experiments.
> The computations can be accelerated through utilizing the pullback geometry. Specifically, each outer product involving the velocity with the Riemannian metric over a small timeframe can be approximated with the corresponding quantity viewed in the output space. For Pullback, this implies the Euclidean distances between the function outputs; for Diet, this implies the spherical distances between the function outputs. Furthermore, the computations along the straight line approximation can be vectorized. As such, they allow reasonably fast evaluations.
>
> **Line 199-200**: we agree that the text can be improved. We stated that it was “meaningful”, as the resulting method was demonstrated to achieve good performances in the experiments on vision foundation models as presented in Section 5, where it consistently yielded the most identifiable relative representations as measured by mean reciprocal ranks.

---

> > ### Comment · Reviewer_age6 · 2025-08-01
> > **Thanks!**
> >
> > I thank the authors for engaging with the comments.
> >
> > Regarding "why it works" I see what you say, but I'm not giving much confidence to the Riemannian interpretation even approximately, since I don't find direct metrics of similarity.. cited works take a very lax modelling perspective in which there is a silent agreement that indirect metrics are enough (which they are not, especially for overparametrized networks!), and in many cases the error bars behind the word "approximately" are taken as rather lax too, as they are just comparative between different approaches and not compared to a theoretical background. Correct me if I'm wrong, please, I'd be relieved to have data to change my mind. That said, this critique however is not stringent in terms of evaluation of this particular paper, since the SOTA is this and the paper compares well to other approaches, as far as I could check.
> >
> > Regarding compute time, I follow the reasoning, but this is not as concrete as I was looking for: what's the experimental real quantification? In your rebuttal you say "reasonably fast evaluations" which is not a metric I understand. Maybe you can stick to the mentioned optimal $N$ and rather than measuring in terms of $|\mathcal A_X|$, plot runtime vs. achieved accuracy (maybe labelling the points with the value of $|\mathcal A_X|$ for completeness) ? And you can add the runtime of the actual model training in the same case, to make your point that stitching is not hard? I see that we are in a time-constrained discussion, so I'm only hoping that you may have saved these data from past training runs (say, from runs underlying mentioned tables 15, 16), in which case I'd like to see them if you can do that.

---

> ### Author Response · Authors · 2025-08-02
> **Response to Reviewer's comment**
>
> We thank the Reviewer for their answer and their engagement. We report below additional answers and results to the points readied by the Reviewer. We hope to have addressed their requests and remain available for further questions.
>
>
> ### **Follow up on "why it works”**
>
> We agree with the Reviewer that a clearer connection between **theoretical results** and **practical methods** for representation alignment is needed. Our intention was to emphasize that the **few theoretical results available** (e.g., \[a]) often rely on **unrealistic assumptions**, such as proofs in **axiomatic settings**, **infinite-data regimes**, or the requirement that **two models learn exactly the same likelihood function**. In our view, a meaningful first step toward bridging theory and practice is to **relax these assumptions**, as initiated in \[b], and begin to **model more realistic scenarios**, e.g. including the **dynamics introduced by model training**.
>
> We believe that **Riemannian methods** can play a key role in this direction to model to capture local alignment beyond linear global transformations of the space (as considered in [a]) and possibly accounting for distortions measured in the linear case in practice. We are also very interested in the **results on overparameterized networks** mentioned by the Reviewer; if a reference can be provided, we would be happy to **add a discussion including this aspect in the manuscript**.
>
> _[a] Roeder, Geoffrey, Luke Metz, and Durk Kingma. *On linear identifiability of learned representations.* ICML, 2021._
>
> _[b] Nielsen, Beatrix MG, et al. *When Does Closeness in Distribution Imply Representational Similarity? An Identifiability Perspective.* arXiv:2506.03784, 2025._
>
> ---
>
>
> ### **Compute Time analysis**
>
>
> We report the running times for two experiments:
>
> 1. **Evaluation on FashionMNIST** (Figure 2) using an **autoencoder**
> 2. **Stitching experiments on CUB** (Figure 14) using Diet decoder and a classification head.
>
> In both cases, we vary the number of anchors used for the relative geodesic representations.
>
> The computational complexity of evaluating **relative geodesic representations** scales with the number of anchors, requiring $N \times |\mathcal{A}_X|$ forward passes of the decoder per data point.
>
> ---
>
> #### **FashionMNIST (Figure 2)**
>
> For FashionMNIST, the decoder is a **convolutional autoencoder** (architecture in **Appendix Table 6**).
> Evaluation is performed on **10k test samples** for both networks, using an **NVIDIA 3080 Ti GPU**.
> The table below reports the number of anchors, evaluation time in seconds, and the corresponding **MRR score**:
>
> | # of Anchors | Time (s) ± Std   | MRR (↑) |
> | ------------ | ---------------- | ------- |
> | 2            | 0.8480 ± 0.0407  | 0.0168  |
> | 3            | 0.8348 ± 0.0367  | 0.0807  |
> | 5            | 0.8377 ± 0.0435  | 0.3503  |
> | 8            | 0.9450 ± 0.0285  | 0.7004  |
> | 10           | 1.0969 ± 0.0297  | 0.8384  |
> | 15           | 1.2853 ± 0.0264  | 0.9296  |
> | 20           | 1.6160 ± 0.0188  | 0.9616  |
> | 25           | 1.8548 ± 0.0218  | 0.9868  |
> | 50           | 3.3311 ± 0.0286  | 0.9981  |
> | 100          | 6.2122 ± 0.0318  | 0.9986  |
> | 300          | 17.6494 ± 0.0653 | 0.9982  |
> | 500          | 29.1925 ± 0.0806 | 0.9986  |
>
> ---
>
> #### **CUB Stitching (Figure 12)**
>
> For the CUB stitching experiments, the decoders are:
>
> * **RelGeo(Pullback):** classification head ( architecture in **Appendix Table 4**)
> * **RelGeo(Diet):** lightweight “diet” decoder (architecture in **Appendix Table 5**)
>
> Experiments are performed on the **CUB dataset** (train/val/test split totaling **11178 samples**) and **averaged across all 16 model pairs**.
> **Note evaluation is faster here due to the shallower decoders compared to the FashionMNIST autoencoder.**
>
> **RelGeo(Pullback)**
>
> | Num Anchors | Time (s) | Accuracy |
> | ----------- | -------- | -------- |
> | 200         | 8.004    | 0.500    |
> | 500         | 21.297   | 0.595    |
> | 1000        | 47.218   | 0.619    |
>
> **RelGeo(Diet)**
>
> | Num Anchors | Time (s) | Accuracy |
> | ----------- | -------- | -------- |
> | 200         | 7.274    | 0.459    |
> | 500         | 19.427   | 0.559    |
> | 1000        | 43.108   | 0.585    |
>
> **Takeaway:** The evaluation of relative geodesics (Equations 4 and 5) is computationally efficient, as confirmed by our measurements. In the **stitching experiments**, the overall runtime is dominated by training the decoders (both the diet decoder and the classifier head). However, this step is also reasonably fast, since we train on only **2,000 samples with augmentations**. For reference, the **total runtime per experiment (including all possible models combinations)** is on the order of **a few hours**. We will include computation times in the manuscript.

---

> > ### Comment · Reviewer_age6 · 2025-08-02
> > **One small thing missing**
> >
> > Thanks for the discussion, what I'm saying is that for overparametrized networks the weights of trained networks necessarily have high variance between seeds, and the optimization finds highly variable parameter values. On the other hand a riemannian manifold with its local linear fixed dimension approximations given by tangent spaces and differentiable charts, seems much less regular than that. This makes me consider the underlying intuition as a "rule of thumb" that works so far, but I wouldn't be surprised if better less regular structures will come out in the future. I don't have a concrete thing to mention, but just to get the idea of what I'm going to, the overall position is partially described in https://arxiv.org/pdf/2507.12224
> >
> > Also keep in mind that what people call "manifold hypothesis" in practice is tested using metrics adapted to fractals and much less regular objects than manifolds: here a paper with an overview for data spaces https://arxiv.org/abs/2207.02862, and I don't know of similar work for actual latent spaces, but I would expect similar irregularities, exacerbated by overparametrization where many parameters might not have a strong pressure to from the random initalization.
> >
> > Please take all the above as some speculation, tangential to the main topic of the paper, and I mention it just because you asked for more detail.
> >
> > About the runtimes, what you gave is good, thank you for taking the time/effort on a Saturday, but there's one **small thing missing**, which maybe I did not state clearly in my previous comment: I'd like that as a comparison, besides the runtimes you gave, you also give (if available) e.g. the runtime for actual training of the models that you stitch. Or some other measure that gives an element of comparison between the times you give, to the scale of complexity of the data/models you are applying it to (as an exaggerated example of what I mean: if I apply any method to some basic MLP on MNIST, it will run faster than if applied to LLAMA types or something bigger, so it might be important to have some other wall-clock quantification related to model complexity, in parallel to runtimes, to compare). I don't know if you have that data, but you can also compare to runtimes of baselines, if available.

---

> > > ### Author Response · Authors · 2025-08-03
> > > **Response to Reviewer comment**
> > >
> > > We thank again the Reviewer for their prompt answers and engagement in discussion.  We answer below. hoping to have addressed the Reviewer's concerns and remain available for further questions.
> > >
> > >
> > > ### **Follow up on why it works**
> > >
> > > We thank the Reviewer for the references, these are highly relevant. We will add the overall discussion to the paper.
> > >
> > >
> > >
> > > ### **Computation Times**
> > >
> > > When reporting computation times for model stitching, we believe it is important to distinguish between the two types of experiments we conduct:
> > >
> > > 1. **Stitching experiments on vision foundation models (Section 5)**
> > >    These experiments are performed on **pretrained backbones**—ViT (Dosovitskiy et al., 2021), DINO v2 (Oquab et al., 2024), and ResNet (He et al., 2016a)—using downstream datasets such as **CUB, CIFAR, ImageNet, and SVHN**.
> > >
> > >    * **Training procedure:** Only a **nonlinear decoder head** (Appendix Table 4) is trained on the downstream dataset.
> > >      * The **backbone pretraining** dominates the overall compute cost and is independent of  the scope of the paper (for example for Dino v2 this  corresponds  on training on **96 A100‑80GB GPUs in parallel** for a total of **3.3 days of training** (Oquab et al., 2024).)
> > >      * For the decoders, the computational cost depends on the size of the dataset used. In our experiments, **training times range from several tens of minutes to roughly 1 hour on a single GPU**. We also note that, in practice, it is often unnecessary to use the entire dataset—especially for large datasets—since the backbone is computationally expensive and the decoder head can typically be adapted efficiently with less samples reaching high accuracy on test set. **We will report precise times for each dataset in the paper**.
> > >
> > >    * **Scalability advantage:**
> > >
> > >      * We gently remark that the cost of stitching **does not scale with model size**, but rather with the **complexity of the transformation between latent spaces** of the models to stitch.
> > >      * Unlike standard adapter-based stitching (Bansal et al., 2021; Csiszárik et al., 2021), which requires ${N \choose 2}$ adapters for $N$ models, our **relative geodesic approach** (following Moschella et al., 2023) uses a **single shared decoder**.
> > >      * Intuitively, instead of learning a separate mapping for every model pair, we map all models into a **canonical relative geodesic space**, enabling **zero-shot stitching** across all pairs.
> > >
> > > 2. **Stitching experiments on autoencoders  (Figures 3 and 4)**
> > >    In this setting, we train small **convolutional autoencoders** from scratch (architectures in Appendix Table 6).
> > >
> > >    * Training each autoencoder is inexpensive—**\~20 minutes** on an RTX 3080 Ti for 50 epochs.
> > >    * **Relative geodesic stitching requires no additional decoder training**, and evaluation is extremely fast—**seconds to under a minute per model pair**, even with hundreds of anchors.
> > >
> > > **Summary:**
> > > Across all settings, **the cost of relative geodesic stitching is negligible compared to model training**. By mapping models to a single canonical space, our method **amortizes the stitching cost** across all pairs and avoids the combinatorial overhead of pairwise adapters.

---

> > > > ### Comment · Reviewer_age6 · 2025-08-03
> > > > **Thanks!**
> > > >
> > > > Ok thanks, this is all what I needed. I surely feel more confident about the paper, and will consider to change my score.

---

> > > > > ### Author Response · Authors · 2025-08-04
> > > > >
> > > > > We thank the Reviewer for their engagement, positive comments and for reconsidering their score.

---

### Official Review · Reviewer_ZLWQ · 2025-07-03

**Clarity:** 3
**Significance:** 2
**Originality:** 3
**Rating:** 4
**Confidence:** 3

**Summary:**

This paper focuses on the similarities of learned representations of vision models on the same task/dataset, either through smaller differences like initialization or larger differences like architecture. Previous work has analyzed this from a linear relative perspective, but this paper takes a geometric analysis and treats the underlying dataset as drawn from an embedded submanifold. Using a Riemannian metric dependent on the encoders/decoders, they propose a Riemannian-metric-aware representation that allows for better stitching and retrieval across features of different models.

**Questions:**

1. Are there reasons why MNIST and CIFAR-10 are enough to demonstrate the need for a Riemannian approach for stitching? Are the datasets themselves not linearly alignable?
2. Would experiments on more modern image datasets like ImageNet run into issues with the linear interpolation scheme?
3. Are there particular benefits to a training-free approach for representation alignment? What does the computation time look like compared to other methods that would require optimization?

**Ethical Concerns:**

["NO or VERY MINOR ethics concerns only"]

**Final Justification:**

The provided experiments and explanation have helped with validating that a Riemannian approach is necessary in these settings and provides nontrivial insight.

**Limitations:**

yes

**Quality:**

2

**Strengths And Weaknesses:**

The work is theoretically well-motivated, building on differential geometry and the pull-back metric formalism, and presents clear derivations and an explicit algorithm (although maybe an equation reference or two would be helpful for the reader to tie back to the paper). The stitching performance increase is notable, showing there is something significant to the authors' method. A Riemannian approach to this representational similarity area is, to my knowledge, still not properly explored.

There are some experimental confusions and concerns. Figure 2 seems to imply that the representation spaces are more or less straight if linear metrics are almost the same as Riemannian metrics, even though I imagine there must be some difference given the notable increase in stitching performance. Still, there is some concern that experiments are restricted to MNIST and CIFAR-10, which are smaller image datasets and potentially more geometrically simple. As stated in limitations, the method assumes that the latent space is convex and thus linear interpolants all lie in a well-defined region for the metric, which may not always be true. In this case, it is still unclear the effectiveness of this versus different variations of a linear alignment. While this paper is a training-free algorithm, it would be nice to compare against linear methods in order to validate the unique usefulness of the geometric approach here.

---

> ### Author Rebuttal · Authors · 2025-07-31
>
> We thank the reviewer for their thoughtful comments. We address their concerns below and remain available for further questions during the discussion period.
>
>
> * **Clarifications about linearity:**
> We respectfully remark that the approximation using the straight line does not correspond to a linear metric.
>
> * **“Figure 2 seems to imply that…”:** Figure 2 and the companion text starting in Line 161 serve as a verification that the proposed straight-line approximation is a meaningful approximation of the ground truth geodesic distance, preserving the rank. We remark that the straight-line approximation corresponds to the Riemannian curve length (or the energy)  of the line (“straight” in the Euclidean sense) under the Riemannian metrics. In other words, both the approximation and the true geodesic use the same Riemannian metric, with the difference lying in the employed curve. The geodesic energy is an upper bound to the true geodesic distance. Additionally we point out that the latter is not scalable to large models, while the approximation nicely scales.
>
> * **Comparison with linear methods:**
> We repeat the stitching experiment of Figure 3, adding two additional baselines:
>     * Linear: fitting a linear map between the two spaces using the anchors
>     * Ortho: fitting an orthogonal map between the two spaces using the anchors, similarly to (Maiorca et al. [2024])
>
>     and report the results in the tables below. We also note that additional qualitative evidence of comparison with linear methods is in Figure 4, fourth column.
>
> **Takeaways:** Our method outperforms both the new baselines on all datasets (MNIST, FMNIST, CIFAR), while being training free. Additionally, we observe that, despite its simplicity, the dataset cannot be aligned with only a few anchors available, as confirmed by our results.
>
>
>
>
>
>
> ### **MNIST**
>
> | Method/Num anchors           | 2               | 3               | 5               | 8               | 10              | 15              | 20              | 25              | 50              | 100             | 300             | 500             |
> | ------------------ | --------------- | --------------- | --------------- | --------------- | --------------- | --------------- | --------------- | --------------- | --------------- | --------------- | --------------- | --------------- |
> | rel cosine         | 0.00 ± 0.00     | 0.02 ± 0.00     | 0.15 ± 0.04     | 0.52 ± 0.05     | 0.68 ± 0.05     | 0.86 ± 0.03     | 0.92 ± 0.01     | 0.94 ± 0.01     | 0.97 ± 0.01     | 0.98 ± 0.00     | 0.99 ± 0.00     | 0.99 ± 0.00     |
> | **rel geo (ours)** | **0.02 ± 0.00** | **0.17 ± 0.05** | **0.85 ± 0.04** | **0.99 ± 0.00** | **1.00 ± 0.00** | **1.00 ± 0.00** | **1.00 ± 0.00** | **1.00 ± 0.00** | **1.00 ± 0.00** | **1.00 ± 0.00** | **1.00 ± 0.00** | **1.00 ± 0.00** |
> | lin                | 0.00 ± 0.00     | 0.00 ± 0.00     | 0.01 ± 0.00     | 0.02 ± 0.00     | 0.03 ± 0.00     | 0.11 ± 0.01     | 0.25 ± 0.02     | 0.43 ± 0.02     | 0.94 ± 0.01     | 1.00 ± 0.00     | 1.00 ± 0.00     | 1.00 ± 0.00     |
> | ortho              | 0.00 ± 0.00     | 0.00 ± 0.00     | 0.01 ± 0.00     | 0.01 ± 0.00     | 0.02 ± 0.00     | 0.09 ± 0.01     | 0.18 ± 0.02     | 0.32 ± 0.03     | 0.90 ± 0.01     | 1.00 ± 0.00     | 1.00 ± 0.00     | 1.00 ± 0.00     |
>
> ---
>
> ### **FMNIST**
>
> | Method/Num anchors                 | 2               | 3               | 5               | 8               | 10              | 15              | 20              | 25              | 50              | 100             | 300             | 500             |
> | ------------------ | --------------- | --------------- | --------------- | --------------- | --------------- | --------------- | --------------- | --------------- | --------------- | --------------- | --------------- | --------------- |
> | rel cosine         | 0.01 ± 0.00     | 0.03 ± 0.01     | 0.17 ± 0.02     | 0.38 ± 0.04     | 0.51 ± 0.03     | 0.68 ± 0.03     | 0.74 ± 0.01     | 0.76 ± 0.03     | 0.83 ± 0.02     | 0.85 ± 0.01     | 0.87 ± 0.01     | 0.88 ± 0.01     |
> | **rel geo (ours)** | **0.02 ± 0.00** | **0.17 ± 0.06** | **0.61 ± 0.14** | **0.78 ± 0.11** | **0.88 ± 0.07** | **0.94 ± 0.02** | **0.96 ± 0.02** | **0.98 ± 0.01** | **0.99 ± 0.00** | **1.00 ± 0.00** | **1.00 ± 0.00** | **1.00 ± 0.00** |
> | lin                | 0.00 ± 0.00     | 0.00 ± 0.00     | 0.01 ± 0.00     | 0.01 ± 0.00     | 0.02 ± 0.00     | 0.05 ± 0.01     | 0.09 ± 0.01     | 0.14 ± 0.01     | 0.54 ± 0.02     | 0.97 ± 0.00     | 1.00 ± 0.00     | 1.00 ± 0.00     |
> | ortho              | 0.00 ± 0.00     | 0.00 ± 0.00     | 0.01 ± 0.00     | 0.01 ± 0.00     | 0.02 ± 0.00     | 0.05 ± 0.01     | 0.08 ± 0.01     | 0.13 ± 0.01     | 0.48 ± 0.02     | 0.95 ± 0.01     | 1.00 ± 0.00     | 1.00 ± 0.00     |
>
> ---
>
> ### **CIFAR**
>
> | Method/Num anchors                 | 2               | 3               | 5               | 8               | 10              | 15              | 20              | 25              | 50              | 100             | 300             | 500             |
> | ------------------ | --------------- | --------------- | --------------- | --------------- | --------------- | --------------- | --------------- | --------------- | --------------- | --------------- | --------------- | --------------- |
> | rel cosine         | 0.00 ± 0.00     | 0.01 ± 0.00     | 0.05 ± 0.02     | 0.12 ± 0.03     | 0.11 ± 0.08     | 0.15 ± 0.10     | 0.19 ± 0.06     | 0.25 ± 0.07     | 0.37 ± 0.06     | 0.32 ± 0.03     | 0.37 ± 0.03     | 0.39 ± 0.03     |
> | **rel geo (ours)** | **0.04 ± 0.00** | **0.39 ± 0.09** | **0.99 ± 0.01** | **1.00 ± 0.00** | **1.00 ± 0.00** | **1.00 ± 0.00** | **1.00 ± 0.00** | **1.00 ± 0.00** | **1.00 ± 0.00** | **1.00 ± 0.00** | **1.00 ± 0.00** | **1.00 ± 0.00** |
> | lin                | 0.00 ± 0.00     | 0.00 ± 0.00     | 0.01 ± 0.00     | 0.03 ± 0.01     | 0.06 ± 0.01     | 0.19 ± 0.03     | 0.35 ± 0.02     | 0.51 ± 0.04     | 0.94 ± 0.01     | 1.00 ± 0.00     | 1.00 ± 0.00     | 1.00 ± 0.00     |
> | ortho              | 0.00 ± 0.00     | 0.00 ± 0.00     | 0.00 ± 0.00     | 0.01 ± 0.00     | 0.01 ± 0.00     | 0.03 ± 0.01     | 0.05 ± 0.02     | 0.07 ± 0.03     | 0.27 ± 0.07     | 0.74 ± 0.08     | 0.98 ± 0.00     | 0.99 ± 0.00     |
>
> * **Experiment on larger datasets:** We provided stitching experiments on vision foundation models in Section 5, which involved larger datasets, demonstrating the performance of our method and of the geodesic energy approximation scheme. We remark that computing true geodesics with models of this size is largely infeasible, due to computational and memory constraints, as well as optimization difficulties since the Jacobian might be ill defined at some points.
>
> * **Benefits to a training-free approach for representation alignment:** For stitching application, having a non parametric method is of fundamental importance for different reasons:
>     - Canonical mapping: Our method can be interpreted as mapping different spaces to a canonical one, invariant to the Riemannian metric adopted (see Figure 1 in the manuscript). This preferred to fitting maps between spaces as the number of adapters to train increases combinatorially with the number of spaces considered. This was observed as well in (Moschella 2023).
>     - Sample efficiency: Our method is particularly sample efficient, requiring few anchors (aligned data) in order to work. As opposed, approaches which involve training (e.g. fitting linear maps or neural networks) require much more aligned data to learn a good map (see for example, the experiment above). This is of fundamental importance for example in multimodal applications, where paired data is costly.

---

> > ### Comment · Reviewer_ZLWQ · 2025-08-07
> >
> > Thank you to the authors for their rebuttal, this has helped clear confusion and the additional experiments are much appreciated.
> >
> > I now understand Fig. 2 and the straight line approximation, thank you for the clarification. While this is necessarily a dense work, NeurIPS is not strictly a geometric venue, so extra care is likely needed for some of these points (maybe giving the straight line approximation a unique name w/ link so the figure is self contained, and a short blurb for why this alignment is important & interesting).
> >
> > I also appreciate the extra experiments provided here and in other rebuttals, which address my presented concerns. I am happy to raise my score.
> >
> > I am still interested in how this geometry looks for more interesting image datasets (doesn't have to be scaled up like ImageNet, e.g. affine MNIST; main idea being that all presented datasets are centered, and affine transformations present nontrivial and rough geometry). But given the additional experiments provided, I am more confident that this paper provides a meaningful claim of Riemannian methods for representation transfer.

---

> > > ### Author Response · Authors · 2025-08-09
> > >
> > > We thank the reviewer for the discussions, the positive comments and for raising their score.

---

### Official Review · Reviewer_7YCq · 2025-07-03

**Clarity:** 1
**Significance:** 2
**Originality:** 3
**Rating:** 3
**Confidence:** 3

**Summary:**

This paper proposes an algorithm that connects two latent structures generated by two different models, using Riemannian geometry of the space of representations. The method aims to grant a Riemannian structure on the space of representations and use them (with anchors) to identify relative geometries of two spaces. This paper verified their algorithm by evaluating reconstruction and stitching capabilities.

**Questions:**

See weaknesses.

**Ethical Concerns:**

["NO or VERY MINOR ethics concerns only"]

**Final Justification:**

After reading the rebuttal and engaging in the discussion, I appreciate the authors’ clarifications. However, I remain concerned that this paper is suggesting two different methods, which don't share many things in common, as their key method. I maintain my score.

**Limitations:**

yes

**Paper Formatting Concerns:**

None.

**Quality:**

2

**Strengths And Weaknesses:**

Strengths
- This paper uses Riemannian metric for identifying the space of representations, and uses it for representation alignment, which looks like an advanced approach in representation alignment methods.

Weaknesses
- (line 122) In the rightmost term of the equation below the line 122, should there be the metric $G_X(x)$ in the middle of the two Jacobians?
- (line 147) Is the $\hat{\gamma}$ the straight line between $z_1$ and $z_2$? What’s the b? Also, if that is a straight line (which will get one temporal parameter – probably $\alpha$ here) it would be better to understand if you write the $\hat{\gamma}$ as a function of $\alpha$.
- Using the energy (or length) of a straight line interpolating two latent points, instead of geodesic distance, looks like a simple heuristic.I understand it’s hard to find any alternatives in a high-dimensional setting like this, but do you have any evidence (theoretical or empirical) that the energy (or length) of a straight line approximates the geodesic distance well?
- The contents in section 3.1 and section 3.2 are pretty much disconnected (e.g. the notation is different – why use the lowercase $f$ for function in section 3.2?). It seems like this paper is proposing two distinct methods (pullback and diet), not a unified single method, representation alignment.
- Section 3.2 is hard to read. Many statements are written without sufficient background knowledge or justification.
  - (line 184) what does it mean that “it can identify the cluster centers of Von-Mises Fisher (VMF) distributions” – e.g. does it mean that the cluster center can be recovered from the latent variables encoded by the sample from the VMF?
  - (line 189) why is it possible to recover the latent variable z by training an encoder f? Under what assumptions does your argument hold?
  - (line 200) Do you mean that the latents are pushed into the unit sphere via a function f, and then the geodesic distance (=cosine distance) is computed?
- In the first experiment (reconstruction), the results of Moschella et al. [1] look visually blurry (figure 4 of this paper, CIFAR10 dataset). However, according to the results in paper [1] (figure 4 of [1]), the reconstruction results look similar to the original ones (visually speaking). What’s the difference?
- In the second experiment (zero-shot stitching) the Imagenet experiment has a better metric with the baseline method than the methods proposed by this paper. Since the imagenet is the largest (and most widely-benchmarked) dataset, should there be a proper explanation about it?

[1] L. Moschella, V. Maiorca, M. Fumero, A. Norelli, F. Locatello, and E. Rodolà. Relative representations enable zero-shot latent space communication. ICLR, 2023, https://openreview.net/forum?id=SrC-nwieGJ.

---

> ### Author Rebuttal · Authors · 2025-07-31
>
> We thank the Reviewer for their detailed comments and feedback. Please find our responses below. We remain available for any additional questions during the discussion period.
>
> * **Typos line 122 and 147:** Thanks for noticing these typos. The convex combination should be:  $\tilde{\gamma}(z\_1,z\_2; \alpha)= (1- \alpha) z\_1 + \alpha z\_2\$. We fixed the typos in the manuscript.
>
> * **Evidence of fidelity of the straight line approximation:** In Figure 2 and the companion text starting in Line 161, we empirically verify that the straight line approximation is generally reasonable by reporting pairwise matrices and showing that the rank is preserved on random pairs of the CIFAR and MNIST dataset. Theoretically this provides an upper bound to the geodesic distance.
>
> * **Pullback and Diet:** We propose the general methodology of using Riemannian distances to enhance relative representations, pulling back a metric \$G\_{\mathcal{X}}\$ from the output space to the latent space. For classifier networks we employ two different choices for \$G\_{\mathcal{X}}\$ and its ambient space: pulling back the Euclidean metric from the logits space (denoted as pullback) and pulling back the spherical metric from the layer before in Diet decoders, assuming a vMF distribution in the space (denoted as Diet). We will clarify the text and improve the flow between Section 3.1 and Section 3.2.
>
> * **“Section 3.2 is hard to read”:** We skipped the technical details to keep the main text concise. We provide a concise summary below, and will provide a more detailed description in the paper. Further technical details can be found in \[1]:
>
>   * Suppose there is a finite set of vectors \$v\_{c}\$ on a unit sphere, representing each class, such that each instance belongs to exactly one class. The latent variables \$z\$ are drawn from a vMF distribution with concentration parameter \$\kappa\$ from the corresponding \$v\_{c}\$, and the data \$x\$ is generated using a function \$g\$ as \$x=g(z)\$. Then, with a Diet decoder, the representations before the final linear head identify the corresponding ground truth \$z\$ up to a transformation.
>   * The recovery of latent variable \$z\$ can be rigorously proven by expanding on the theoretical framework of non-linear ICA. The theory holds under reasonable assumptions on the data generation process and the embeddings and weights in Diet decoder’s final layer. For a formal treatment of the assumptions, we refer the reviewer to Assumptions 1C and Theorem 1C in \[1].
>   * Theory suggests that a trained Diet decoder is capable of identifying the ground truth \$z\$ up to a transformation. As such, we directly assign a spherical geometry to \$z\$, and compute the distances accordingly.
>
> * **Reconstruction experiment (stitching):** In the experiment in Figure 4 we consider a setting where the number of anchors is limited, i.e., 5 for MNIST and FMNIST datasets and 10 for CIFAR, which leads to the difference: in the corresponding Figure 4 in (Moschella et al. 2023), the number of anchors was set to 500 (cfr. Appendix A.5.5 in Moschella et al. 2023). This highlights the huge increase in sample efficiency of our approach, as confirmed by our results in Figure 3.
>
> * **Performance on Imagenet (stitching):** We currently do not have a definitive explanation of this, and it remains an interesting avenue for future work.
>
> \[1] *Cross-Entropy Is All You Need To Invert the Data Generating Process*, Reizinger et al., ICLR 2025

---

> > ### Comment · Reviewer_7YCq · 2025-08-05
> >
> > Thank you for your response. It helped me understand the method much better and some of my issues are addressed. However, I still have concerns on the discrepancies between two proposed methods, pullback and Diet.
> >
> > The pullback method grants Riemannian metric on representation space by pulling back the output space (reconstruction for autoencoder, logits for classifier). It then approximates geodesic distances by numerically computing integrals along linear interpolation.
> >
> > On the other side, the Diet method pushes the representation space into a unit sphere and measures cosine distance directly. Although the cosine distance could be seen as a Riemannian metric on the unit sphere, the two approaches otherwise have little in common.
> >
> > This difference is reflected in the experimental results: Diet is better for matching reconstruction, while the pullback is better for classification. If no single method works well across tasks, the approach may be less suitable for diverse applications.

---

> ### Author Response · Authors · 2025-08-05
>
> We thank the Reviewer for their response. We address below their additional questions and remain available for any further clarification.
>
> ### **Diet vs. Pullback**
>
> We would like to gently clarify that **both “Pullback” and “Diet” are proper pullback metrics**:
>
> "Pullback" uses the Euclidean metric from the classifier’s logit space (or from the autoencoder output space), as the Reviewer correctly noted.
>
> **Diet uses the spherical (cosine) metric pulled back from the penultimate layer of the Diet decoder**.
>
> **Both metrics compute the Riemannian arc length (or energy) by numerically integrating along linear interpolations in the latent space**. In practice, Diet also relies on a discretization along the interpolation path.
>
> In our experiments, we typically use 15–30  dicretization steps, so the method effectively integrates the pullback metric along the path.
>
> In summary, the two metrics differ in:
>
> 1. The metric being pulled back (Euclidean vs. cosine).
>
> 2. The output space considered (logit/output space vs. Diet penultimate layer).
>
> These choices impact downstream tasks: for instance, Diet is more suited for retrieval tasks, while Pullback performs better for stitching, as the Reviewer observes as well. It is in general not clear if a single metric can be optimal for all tasks.
>
> Finally, we note that **different pullback metrics can be combined**, either by constructing product metrics (via concatenation) or by interpolating between metrics, which we see as a promising future direction to get good perfomance on multiple diverse tasks with a **unique representation**.

---

> > ### Comment · Reviewer_7YCq · 2025-08-08
> >
> > Thank you for your detailed response, which helped clarify some aspects of the methods. However, the discrepancy between the two proposed approaches—Pullback and Diet—remains a significant concern. While the clarification that both methods are pullback metrics is helpful, the differences in their performance across tasks (e.g., Diet excelling in retrieval and Pullback in classification) suggest that no single method is universally effective. This limits the general applicability of the approach. Additionally, the potential for combining metrics is promising but remains an open problem. Given these unresolved issues, I maintain my score.

---

> ### Author Response · Authors · 2025-08-08
>
> We thank the reviewer for their answer and engagement. We would like to gently remark that stitching and retrieval are to some extent orthogonal tasks, as the former depends on identifiability up to class and the later up to instance, as such it makes sense that there is not a single metric outperforming on every downstream tasks. We also gently remark that:
>
> * Diet still performs well and remains competitive on stitching, though may not be *excelling*,
> * For practitioners it is generally known beforehand whether the task concerns stitching or retrieval. As such, choosing a right metric for the downstream task generally does not affect the generality of the method.
>
> We remain available for any further clarifications.

---

### Note · Authors · 2025-08-15

We thank the Reviewers for their suggestions and engaging discussions, and the AC for ensuring a smooth, fair review process.

We are encouraged that our formulation was valued from both **theoretical** and **practical** perspectives:

* **Theoretical:** *“the work is theoretically well-motivated”* (ZLWQ), *“solid formulation grounded in differential geometry”* (k3Ho)
* **Practical:** *“the stitching performance increase is notable”* (ZLWQ), *“the strength of the paper is that the experiments work”* (age6)

During the rebuttal, we:

* Clarified formulations to Reviewers **7YCq** and **ZLWQ**
* Provided empirical results to **ZLWQ**, **age6**, and **k3Ho**, including multiple modalities, new baselines, anchor-sampling ablations, and computation-time analysis

We are pleased that **Reviewers age6 and k3Ho reconsidered their ratings** and **Reviewer ZLWQ raised their score**. Reviewer **7YCq** maintained some concerns on Pullback vs Diet, which we addressed in detail.

In our work, we introduced a **Riemannian geometric framework** for connecting latent spaces of independently trained neural networks.

1. **Motivation**
   * The **relative representations** framework \[1] aligns latent spaces but ignores geometry, despite the hypothesis that real-world data lie on **low-dimensional manifolds**.
   * In Riemannian geometry, **geodesic distances** are preserved under isometric transformations, providing stronger invariance guarantees.

2. **Relative Geodesic Representations**
   * Use **Riemannian curve lengths** as features.
   * Explore two Riemannian structures:

     * **Pullback metric** from the Euclidean output space
     * **Diet pullback metric** from a decoder trained with a spherical metric (with identifiability guarantees)
   * More generally, our method can pull back **any metric from an output space, leveraging geometry for the downstream task**.

3. **Key results**
   * **Autoencoders:** Pullback metric improves performance, especially with few anchors.
   * **Vision foundation models:** Pullback metric excels in stitching; Diet metric excels in retrieval.

**In summary**, **Riemannian geometry enhances relative representations**, enabling accurate stitching and retrieval, and showing that distinct neural networks can learn the **same underlying manifold**. This opens for future work in representation alignment from **theoretical** (identifiability) and **practical** (model merging and reuse) perspectives.

Sincerely,

Authors of submission 27217

---

### Decision · Program_Chairs · 2025-09-17

**Decision:**

Accept (poster)

**Comment:**

This paper leverages Riemannian geometry to build representations from pullback metrics. The proposed ideas allow to align representations learned by different networks, e.g. by encoding a datapoint with one autoencoder, aligning the representation with the latent space of a different autoencoder, and then decoding with the second decoder. The high degree to which the authors show different latent spaces can be aligned is quite surprising, and provides strong evidence suggesting different models are really learning the same underlying manifold. I believe the community would benefit from being aware of these findings.

The empirical results are particularly strong and were praised by the reviewers. The paper received split reviews, with two reviewers recommending borderline rejection, one reviewer recommending borderline acceptance, and one reviewer strongly championing the paper. The main criticism from the reviewers who recommended rejection is that this is a dense paper, and that it will likely be hard to follow for readers who do not have a background in differential geometry. The other two reviewers found the paper to be well written and clear. I went through the paper carefully and while I agree that readers without a background in geometry might struggle to understand some aspects of the paper, I also believe this is a clear and well written paper for its intended audience.

Overall, this paper presents a well-motivated approach that obtains strong empirical results. I believe this paper is valuable to the community working on leveraging tools from differential geometry for machine learning, and that the difficulty of the material to those lacking background in geometry is not a compelling reason for rejection. I thus recommend acceptance. This being said, with the goal of making the paper more accessible to readers without a background in differential geometry, I also ask the authors to please add a more complete informal description of the method for the camera-ready version of the paper.